# SPEAR: Exact Gradient Inversion of Batches in Federated Learning

**Dimitar I. Dimitrov**[1], **Maximilian Baader**[2], **Mark Niklas Müller**[2,3], **Martin Vechev**[2]

[1] INSAIT, Sofia University "St. Kliment Ohridski"     [2] ETH Zurich     [3] LogicStar.ai

{dimitar.iliev.dimitrov}@insait.ai [1]

{mbaader, mark.mueller, martin.vechev}@inf.ethz.ch [2]

## Abstract

Federated learning is a framework for collaborative machine learning where clients only share gradient updates and not their private data with a server. However, it was recently shown that gradient inversion attacks can reconstruct this data from the shared gradients. In the important honest-but-curious setting, existing attacks enable exact reconstruction only for batch size of $b = 1$, with larger batches permitting only approximate reconstruction. In this work, we propose SPEAR, *the first algorithm reconstructing whole batches with $b > 1$ exactly*. SPEAR combines insights into the explicit low-rank structure of gradients with a sampling-based algorithm. Crucially, we leverage ReLU-induced gradient sparsity to precisely filter out large numbers of incorrect samples, making a final reconstruction step tractable. We provide an efficient GPU implementation for fully connected networks and show that it recovers high-dimensional ImageNet inputs in batches of up to $b \lesssim 25$ exactly while scaling to large networks. Finally, we show theoretically that much larger batches can be reconstructed with high probability given exponential time.

## 1 Introduction

**Federated Learning** has emerged as the dominant paradigm for training machine learning models collaboratively without sharing sensitive data [2]. Instead, a central server sends the current model to all clients which then send back gradients computed on their private data. The server aggregates the gradients and uses them to update the model. Using this approach sensitive data never leaves the clients' machines, aligning it better with data privacy regulations such as the General Data Protection Regulation (GDPR) and California Consumer Privacy Act (CCPA).

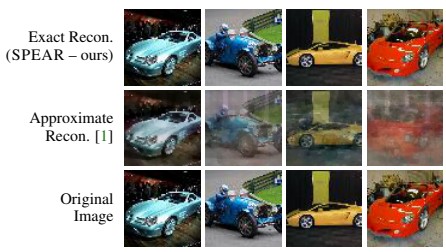

Figure 1: A sample of four images from a batch of $b = 20$, reconstructed using our SPEAR (top) or the prior state-of-the-art Geiping et al. [1] (mid), compared to the ground truth (bottom).

**Gradient Inversion Attacks** Recent work has shown that an honest-but-curious server can use the shared gradient updates to recover the sensitive client data [3, 4]. However, while *exact* reconstruction was shown to be possible for batch sizes of $b = 1$ [5, 6], it was assumed to be infeasible for larger batches. This led to a line of research on approximate methods that sacrificed reconstruction quality in order to recover batches of $b > 1$ inputs [7, 8, 9]. In this paper we challenge this fundamental assumption and, for the first time, show that exact reconstruction is possible for batch sizes $b > 1$.

**This Work: Exact Reconstruction of Batches** We propose the *first gradient inversion attack reconstructing inputs exactly for batch sizes $b > 1$* in the honest-but-curious setting. In Fig. 1, we show the resulting reconstructions versus approximate methods [1] for a batch of $b = 20$ images.

38th Conference on Neural Information Processing Systems (NeurIPS 2024).

Our approach leverages two key properties of gradient updates in fully connected ReLU networks: First, these gradients have a specific *low-rank structure* due to small batch sizes $b \ll n, m$ compared to the input dimensionality $n$ and the hidden dimension $m$. Second, the (unknown) gradients with respect to the inputs of the first ReLU layer are sparse due to the ReLU function itself. We combine these properties with ideas from sparsely-used dictionary learning [10] to propose a sampling-based algorithm, called SPEAR (**Sp**arsity **E**xploiting **A**ctivation **R**ecovery) and show that it succeeds with high probability for $b < m$. While SPEAR scales exponentially with batch size $b$, we provide a highly parallelized GPU implementation, which empirically allows us to reconstruct batches of size up to $b \lesssim 25$ exactly even for large inputs (IMAGENET) and networks (widths up to 2000 neurons and depths up to 9 layers) in around one minute per batch.

**Main Contributions:**
- The first gradient inversion attack showing theoretically that *exact reconstruction* of complete batches with size $b > 1$ in the honest-but-curious setting is possible.
- SPEAR: a sampling-based algorithm leveraging *low rankness* and ReLU-induced *sparsity of gradients* for exact gradient inversion that succeeds with high probability.
- A highly parallelized GPU implementation of SPEAR, which we empirically demonstrate to be effective across a wide range of settings and make publicly available on GitHub.

## 2 Method Overview

We first introduce our setting before giving a high-level overview of our attack SPEAR, whose sketch is shown in Fig. 2.

**Setting** We consider a neural network $f$ containing a linear layer $z = Wx + b$ followed by ReLU activations $y = \text{ReLU}(z)$ trained with a loss function $\mathcal{L}$. Let now $X \in \mathbb{R}^{n \times b}$ be a batch of $b$ inputs to the linear layer $Z = WX + (b| \ldots |b)$, with weights $W \in \mathbb{R}^{m \times n}$, bias $b \in \mathbb{R}^m$ and output $Z \in \mathbb{R}^{m \times b}$. Further, let $Y \in \mathbb{R}^{m \times b}$ be the result of applying the ReLU activation to $Z$, i.e., $Y = \text{ReLU}(Z)$ and assume $b \leq m, n$. The goal of SPEAR is to *recover the inputs $X$* (up to permutation) given the gradients $\frac{\partial \mathcal{L}}{\partial W}$ and $\frac{\partial \mathcal{L}}{\partial b}$ (see Fig. 2, i).

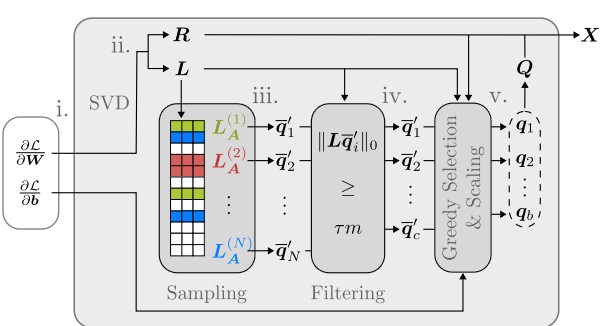

Figure 2: Overview of SPEAR. The gradient $\frac{\partial \mathcal{L}}{\partial W}$ is decomposed to $R$ and $L$. Sampling gives $N$ proposal directions, which we filter down to $c$ candidates via a sparsity criterion with threshold $\tau * m$. A greedy selection method selects batchsize $b$ directions. Scale recovery via $\frac{\partial \mathcal{L}}{\partial b}$ returns the disaggregation matrix $Q$ and thus the inputs $X$.

**Low-Rank Decomposition** We first show that the weight gradient $\frac{\partial \mathcal{L}}{\partial W} = \frac{\partial \mathcal{L}}{\partial Z} X^\top$ naturally has a low rank $b \leq m, n$ (Theorem 3.1) and can therefore be decomposed as $\frac{\partial \mathcal{L}}{\partial W} = LR$ with $L \in \mathbb{R}^{m \times b}$ and $R \in \mathbb{R}^{b \times n}$ using SVD (Fig. 2, ii). We then prove the existence disaggregation matrix $Q = (q_1| \ldots |q_b) \in \text{GL}_b(\mathbb{R})$, allowing us to express the inputs as $X^\top = Q^{-1} R$ and activation gradients as $\frac{\partial \mathcal{L}}{\partial Z} = LQ$ (Theorem 3.2). Next, we leverages the sparsity of $\frac{\partial \mathcal{L}}{\partial Z}$ to recover $Q$ exactly.

**ReLU Induced Sparsity** We show that ReLU layers induce sparse activation gradients $\frac{\partial \mathcal{L}}{\partial Z}$ (Sec. 3.2). We then leverage this sparsity to show that, with high probability, there exist submatrices $L_A \in \mathbb{R}^{b-1 \times b}$ of $L$, such that their kernel is an unscaled column $\overline{q}_i$ of our disaggregation matrix $Q$, i.e., $\ker(L_A) = \text{span}(q_i)$, for all $i \in \{1, \ldots, b\}$ (Theorem 3.3). Given these unscaled colmuns $\overline{q}_i$, we recover their scale by leveraging the bias gradient $\frac{\partial \mathcal{L}}{\partial b}$ (Theorem 3.5).

**Sampling and Filtering Directions** To identify the submatrices $L_A$ of $L$ which induce the directions $\overline{q}_i$, we propose a sampling approach (Sec. 4.1): We randomly sample $b - 1$ rows of $L$ to obtain an $L_A$ and thus proposal direction $\overline{q}_i' = \ker(L_A)$ (Fig. 2 iii). Crucially, the product $L\overline{q}_i' = \frac{\partial \mathcal{L}}{\partial z_i}$ recovers a column of the sparse activation gradient $\frac{\partial \mathcal{L}}{\partial Z}$ for correct directions $\overline{q}_i'$ and a dense linear combination of such columns for incorrect ones. This sparsity gap allows the large number $N$ of proposal directions obtained from submatrices $L_A$ to be filtered to $c \gtrsim b$ unique candidates (Fig. 2 iv).

**Greedy Direction Selection** We now have to select the correct $b$ directions from our set of $c$ candidates (Fig. 2, v). To this end, we build an initial solution $\boldsymbol{Q}'$ from the $b$ directions inducing the highest sparsity in $\frac{\partial \mathcal{L}}{\partial \boldsymbol{Z}}' = \boldsymbol{L}\boldsymbol{Q}'$. To assess the quality of this solution $\boldsymbol{Q}'$, we introduce the sparsity matching score $\sigma$ which measures how well the sparsity of the activation gradients $\frac{\partial \mathcal{L}}{\partial \boldsymbol{Z}}'$ matches the ReLU activation pattern induced by the reconstructed input $\boldsymbol{X}'^{\top} = \boldsymbol{Q}'^{-1}\boldsymbol{R}$. Finally, we greedily optimize $\boldsymbol{Q}'$ to maximize the sparsity matching score, by iteratively replacing an element $\boldsymbol{q}'_i$ of $\boldsymbol{Q}'$ with the candidate direction $\boldsymbol{q}'_j$ yielding the greatest improvement in $\sigma$ until convergence. We can then validate the resulting input $\boldsymbol{X}^{\top} = \boldsymbol{Q}^{-1}\boldsymbol{R}$ by checking whether it induces the correct gradients. We formalize this as Alg. 1 in Sec. 5 and show that it succeeds with high probability for $b < m$.

## 3 Gradient Inversion via Sparsity and Low-Rankness

In this section, we will demonstrate that both low rankness and sparsity arise naturally for gradients of fully connected ReLU networks and explain theoretically how we recover $\boldsymbol{X}$. Specifically, in Sec. 3.1, we first argue that $\frac{\partial \mathcal{L}}{\partial \boldsymbol{W}} = \frac{\partial \mathcal{L}}{\partial \boldsymbol{Z}}\boldsymbol{X}^T$ follows direclty from the chain rule. We then show that for every decomposition $\frac{\partial \mathcal{L}}{\partial \boldsymbol{W}} = \boldsymbol{L}\boldsymbol{R}$, there exists an unknown disaggregation matrix $\boldsymbol{Q}$ allowing us to reconstruct $\boldsymbol{X}^{\top} = \boldsymbol{Q}^{-1}\boldsymbol{R}$ and $\frac{\partial \mathcal{L}}{\partial \boldsymbol{Z}} = \boldsymbol{L}\boldsymbol{Q}$. The remainder of the section then focuses on recovering $\boldsymbol{Q}$. To this end, we show in Sec. 3.2 that ReLU layers induce sparsity in $\frac{\partial \mathcal{L}}{\partial \boldsymbol{Z}}$, which we then leveraged in Sec. 3.3 to reconstruct the columns of $\boldsymbol{Q}$ up to scale. Finally, in Sec. 3.4, we show how the scale of $\boldsymbol{Q}$'s columns can be recovered from $\frac{\partial \mathcal{L}}{\partial \boldsymbol{b}}$. Unless otherwise noted, we defer all proofs to App. B.

### 3.1 Explicit Low-Rank Representation of $\frac{\partial \mathcal{L}}{\partial \boldsymbol{W}}$

We first show that the weight gradients $\frac{\partial \mathcal{L}}{\partial \boldsymbol{W}}$ can be written as follows:

**Theorem 3.1.** *The network's gradient w.r.t. the weights $\boldsymbol{W}$ can be represented as the matrix product:*

$$\frac{\partial \mathcal{L}}{\partial \boldsymbol{W}} = \frac{\partial \mathcal{L}}{\partial \boldsymbol{Z}}\boldsymbol{X}^T. \tag{1}$$

For batch sizes $b \leq n, m$, the dimensionalities of $\frac{\partial \mathcal{L}}{\partial \boldsymbol{Z}} \in \mathbb{R}^{m \times b}$ and $\boldsymbol{X} \in \mathbb{R}^{n \times b}$ in Eq. 1 directly yield that the rank of $\frac{\partial \mathcal{L}}{\partial \boldsymbol{W}}$ is at most $b$. This confirms the observations of Kariyappa et al. [9] and shows that $\boldsymbol{X}$ and $\frac{\partial \mathcal{L}}{\partial \boldsymbol{Z}}$ correspond to a specific low-rank decomposition of $\frac{\partial \mathcal{L}}{\partial \boldsymbol{W}}$.

To actually find this decomposition and thus recover $\boldsymbol{X}$, we first consider an arbitrary decomposition of the form $\frac{\partial \mathcal{L}}{\partial \boldsymbol{W}} = \boldsymbol{L}\boldsymbol{R}$, where $\boldsymbol{L} \in \mathbb{R}^{m \times b}$ and $\boldsymbol{R} \in \mathbb{R}^{b \times n}$ are of maximal rank. We chose the decomposition obtained via the reduced SVD decomposition of $\frac{\partial \mathcal{L}}{\partial \boldsymbol{W}} = \boldsymbol{U}\boldsymbol{S}\boldsymbol{V}$ by setting $\boldsymbol{L} = \boldsymbol{U}\boldsymbol{S}^{\frac{1}{2}}$ and $\boldsymbol{R} = \boldsymbol{S}^{\frac{1}{2}}\boldsymbol{V}$, where $\boldsymbol{U} \in \mathbb{R}^{n \times b}$, $\boldsymbol{S} \in \mathbb{R}^{b \times b}$ and $\boldsymbol{V} \in \mathbb{R}^{b \times n}$. We now show that there exists an unique disaggregation matrix $\boldsymbol{Q}$ recovering $\boldsymbol{X}$ and $\frac{\partial \mathcal{L}}{\partial \boldsymbol{Z}}$ from $\boldsymbol{L}$ and $\boldsymbol{R}$:

**Theorem 3.2.** *If the gradient $\frac{\partial \mathcal{L}}{\partial \boldsymbol{Z}}$ and the input matrix $\boldsymbol{X}$ are of full-rank and $b \leq n, m$, then there exists an unique matrix $\boldsymbol{Q} \in \mathbb{R}^{b \times b}$ of full-rank s.t. $\frac{\partial \mathcal{L}}{\partial \boldsymbol{Z}} = \boldsymbol{L}\boldsymbol{Q}$ and $\boldsymbol{X}^T = \boldsymbol{Q}^{-1}\boldsymbol{R}$.*

Theorem 3.2 is a direct application of Lemma B.1 shown in App. B, a general linear algebra result stating that under most circumstances different low-rank matrix decompositions can be transformed into each other via an unique invertible matrix. Crucially, this implies that recovering the input $\boldsymbol{X}$ and the gradient $\frac{\partial \mathcal{L}}{\partial \boldsymbol{Z}}$ matrices is equivalent to obtaining the unique disaggregation matrix $\boldsymbol{Q}$. Next, we show how the ReLU-induced sparsity patterns in $\frac{\partial \mathcal{L}}{\partial \boldsymbol{Z}}$ or $\boldsymbol{X}$ can be leveraged to recover $\boldsymbol{Q}$ exactly.

### 3.2 ReLU-Induced Sparsity

ReLU activation layers can induce sparsity both in the gradient $\frac{\partial \mathcal{L}}{\partial \boldsymbol{Z}}$ (if the ReLU activation succeeds the considered linear layer) or in the input (if the ReLU activation precedes the linear layer).

**Gradien Sparsity** If a ReLU activation succeeds the linear layer, i.e., $\boldsymbol{Y} = \text{ReLU}(\boldsymbol{Z})$, we have $\frac{\partial \mathcal{L}}{\partial \boldsymbol{Z}} = \frac{\partial \mathcal{L}}{\partial \boldsymbol{Y}} \odot \mathbb{1}_{[\boldsymbol{Z}>0]}$, where $\odot$ is the elementwise multiplication and $\mathbb{1}_{[\boldsymbol{Z}>0]}$ is a matrix of 0s and 1s with each entry indicating if the corresponding entry in $\boldsymbol{Z}$ is positive. At initialization, roughly half of the entries in $\boldsymbol{Z}$ are positive, making $\frac{\partial \mathcal{L}}{\partial \boldsymbol{Z}}$ sparse with $\sim 0.5$ of the entries $= 0$.

**Input Sparsity**    ReLUs also introduce sparsity if the linear layer in question is preceded by a ReLU activation. Here, $\boldsymbol{X} = \mathrm{ReLU}(\tilde{\boldsymbol{Z}})$ will again be sparse with $\sim 0.5$ of the entries $= 0$ at initialization.

Note that for all but the first and the last layer of a fully connected network, we have sparsity in both, $\boldsymbol{X}$ and $\frac{\partial \mathcal{L}}{\partial \boldsymbol{Z}}$. Due to the symmetry of their formulas in Theorem 3.2, our method can be applied in all three arising sparsity settings. In the remainder of this work, we assume w.l.o.g. that only $\frac{\partial \mathcal{L}}{\partial \boldsymbol{Z}}$ is sparse, corresponding to the first layer of a fully connected network. We now describe how to leverage this sparsity to compute the disaggregation matrix $\boldsymbol{Q}$ and thus recover the input batch $\boldsymbol{X}$.

### 3.3   Breaking Aggregation through Sparsity

Our exact recovery algorithm for the disaggregation matrix $\boldsymbol{Q}$ is based on the following insight:

If we can construct two submatrices $\boldsymbol{A} \in \mathbb{R}^{b-1 \times b}$ and $\boldsymbol{L}_A \in \mathbb{R}^{b-1 \times b}$ by choosing $b-1$ rows with the same indices from $\frac{\partial \mathcal{L}}{\partial \boldsymbol{Z}}$ and $\boldsymbol{L}$, respectively, such that $\boldsymbol{A}$ has full rank and an all-zero $i^{\text{th}}$ column, then the kernel $\ker(\boldsymbol{L}_A)$ of $\boldsymbol{L}_A$ contains a column $\boldsymbol{q}_i$ of $\boldsymbol{Q}$ up to scale. We formalize this as follows:

**Theorem 3.3.** *Let $\boldsymbol{A} \in \mathbb{R}^{b-1 \times b}$ be a submatrix of $\frac{\partial \mathcal{L}}{\partial \boldsymbol{Z}}$ s.t. its $i^{\text{th}}$ column is $\boldsymbol{0}$ for some $i \in \{1, \ldots, b\}$. Further, let $\frac{\partial \mathcal{L}}{\partial \boldsymbol{Z}}$, $\boldsymbol{X}$, and $\boldsymbol{A}$ be of full rank and $\boldsymbol{Q}$ be as in Theorem 3.2. Then, there exists a full-rank submatrix $\boldsymbol{L}_A \in \mathbb{R}^{b-1 \times b}$ of $\boldsymbol{L}$ s.t. $\mathrm{span}(\boldsymbol{q}_i) = \ker(\boldsymbol{L}_A)$ for the $i^{\text{th}}$ column $\boldsymbol{q}_i$ of $\boldsymbol{Q} = (\boldsymbol{q}_1 | \cdots | \boldsymbol{q}_b)$.*

*Proof.* Pick an $i \in \{1, \ldots, b\}$. By assumption, there exists a submatrix $\boldsymbol{A} \in \mathbb{R}^{b-1 \times b}$ of $\frac{\partial \mathcal{L}}{\partial \boldsymbol{Z}}$ of rank $b-1$ whose $i^{\text{th}}$ column is $\boldsymbol{0}$. To construct $\boldsymbol{L}_A$, we take rows from $\boldsymbol{L}$ with indices corresponding to $\boldsymbol{A}$'s row indices in $\frac{\partial \mathcal{L}}{\partial \boldsymbol{Z}}$. As $\frac{\partial \mathcal{L}}{\partial \boldsymbol{Z}}$ and $\boldsymbol{X}$ have full rank, by Theorem 3.2, we know that $\frac{\partial \mathcal{L}}{\partial \boldsymbol{Z}} = \boldsymbol{L}\boldsymbol{Q}$, and hence $\boldsymbol{A} = \boldsymbol{L}_A \boldsymbol{Q}$. Multiplying from the right with $\boldsymbol{e}_i$ yields $0 = \boldsymbol{A}\boldsymbol{e}_i = \boldsymbol{L}_A \boldsymbol{Q} \boldsymbol{e}_i = \boldsymbol{L}_A \boldsymbol{q}_i$, and hence $\ker(\boldsymbol{L}_A) \supseteq \mathrm{span}(\boldsymbol{q}_i)$. Further, as $\mathrm{rank}(\boldsymbol{A}) = b-1$ and $\mathrm{rank}(\boldsymbol{Q}) = b$, we have that $\mathrm{rank}(\boldsymbol{L}_A) = b-1$. By the rank-nullity theorem $\dim(\ker(\boldsymbol{L}_A)) = 1$ and hence $\ker(\boldsymbol{L}_A) = \mathrm{span}(\boldsymbol{q}_i)$. $\qquad\square$

As $\frac{\partial \mathcal{L}}{\partial \boldsymbol{Z}}$ is not known a priori, we can not simply search for such a set of rows. Instead, we have to sample submatrices $\boldsymbol{L}_A$ of $\boldsymbol{L}$ at random and then filter them using the approach discussed in Sec. 4. However, we will show in Sec. 5.2 that we will find suitable submatrices with high probability for $b < m$ due to the sparsity of $\frac{\partial \mathcal{L}}{\partial \boldsymbol{Z}}$ and the large number $\binom{m}{b-1}$ of possible submatrices. We will now discuss how to recover the scale of the columns $\boldsymbol{q}_i$ given their unscaled directions $\overline{\boldsymbol{q}}_i$ forming $\overline{\boldsymbol{Q}}$.

### 3.4   Obtaining $Q$: Recovering the Scale of columns in $\overline{Q}$

Given a set of $b$ correct directions $\overline{\boldsymbol{Q}} = (\overline{\boldsymbol{q}_1} | \cdots | \overline{\boldsymbol{q}_b})$, we can recover their scale, enabling us to reconstruct $\boldsymbol{X}$, as follows. We first represent the correctly scaled columns as $\boldsymbol{q}_i = s_i \cdot \overline{\boldsymbol{q}}_i$ with the unknown scale parameters $s_i \in \mathbb{R}$. Now, recovering the scale is equivalent to computing all $s_i$. To this end, we leverage the gradient w.r.t. the bias $\frac{\partial \mathcal{L}}{\partial \boldsymbol{b}}$:

**Theorem 3.4.** *The gradient w.r.t. the bias $\boldsymbol{b}$ can be written in the form $\frac{\partial \mathcal{L}}{\partial \boldsymbol{b}} = \frac{\partial \mathcal{L}}{\partial \boldsymbol{Z}} \begin{bmatrix} 1 \\ \vdots \\ 1 \end{bmatrix}$.*

Thus, the coefficients $s_i$ can be calculated as:

**Theorem 3.5.** *For any left inverse $\boldsymbol{L}^{-L}$ of $\boldsymbol{L}$, we have $\begin{bmatrix} s_1 \\ \vdots \\ s_b \end{bmatrix} = \overline{\boldsymbol{Q}}^{-1} \boldsymbol{L}^{-L} \frac{\partial \mathcal{L}}{\partial \boldsymbol{b}}$*

Theorem 3.5 allows us to directly obtain the true matrix $\boldsymbol{Q} = \overline{\boldsymbol{Q}} \, \mathrm{diag}(s_1, \ldots, s_b)$ from the unscaled matrix $\overline{\boldsymbol{Q}}$. We now discuss how to recover $\overline{\boldsymbol{Q}}$ via sampling and filtering candidate directions $\overline{\boldsymbol{q}}_i$.

## 4   Efficient Filtering and Validation of Candidates

In the previous section, we saw that given the correct selection of submatrices $\boldsymbol{L}_A$, we can recover $\boldsymbol{Q}$ directly. However, we do not know how to pick $\boldsymbol{L}_A$ a priori. To solve this, we rely on a sampling approach: We first randomly sample submatrices $\boldsymbol{L}_A$ of $\boldsymbol{L}$ and corresponding direction candidates $\overline{\boldsymbol{q}}'$ spanning $\ker(\boldsymbol{L}_A)$. However, checking whether $\overline{\boldsymbol{q}}'$ is a valid direction is not straightforward as we do not know $\frac{\partial \mathcal{L}}{\partial \boldsymbol{Z}}$ and hence can not observe $\boldsymbol{A}$ directly as reconstructing $\frac{\partial \mathcal{L}}{\partial \boldsymbol{Z}} = \boldsymbol{L}\boldsymbol{Q}$ requires the full $\boldsymbol{Q}$.

To address this, we filter the majority of wrong proposals $\overline{q}'$ using deduplication and a sparsity-based criterion (Sec. 4.1), leaving us with a set of candidate directions $\mathcal{C} = \{\overline{q}'_j\}_{j \in \{1,\dots,c\}}$. We then select the correct directions in $\mathcal{C}$ greedily based on a novel sparsity matching score (Sec. 4.2).

## 4.1 Efficient Filtering of Directions $\overline{q}'$

**Filtering Mixtures via Sparsity** It is highly likely ($p = (1 - \frac{1}{2^{b-1}})^b$) that a random submatrix of $L$ will not correspond to an $A$ with any $0$ column. We filter these directions by leveraging the following insight. The kernel of such submatrices is spanned by a linear combination $\overline{q}' = \sum_i \alpha_i \overline{q}_i$. Thus $L\overline{q}'$ will be a linear combination of sparse columns of $\frac{\partial \mathcal{L}}{\partial Z}$. As this sparsity structure is random, linear combinations will have much lower sparsity with high probability. We thus discard all candidates $\overline{q}'$ with sparsity of $L\overline{q}'$ below a threshold $\tau$, chosen to make the probability of falsely rejecting a correct direction $p_{fr}(\tau, m) = \frac{1}{2^m} \sum_{i=0}^{\lfloor m \cdot \tau \rfloor} \binom{m}{i}$, obtained from the cumulative distribution function of the binomial distribution, small. For example for $m = 400$ and $p_{fr}(\tau, m) < 10^{-5}$, we have $\tau = 0.395$. We obtain the candidate pool $\mathcal{C} = \{\overline{q}'_j\}_{j \in \{1,\dots,c\}}$ from all samples that were not filered this way.

**Filtering Duplicates** As it is highly likely to have multiple full-rank submatrices $A$, whose $i^{\text{th}}$ column is $0$, we expect to sample the same proposal $\overline{q}'_i$ multiple times. We remove these duplicates to substantially reduce our search space.

## 4.2 Greedy Optimization

While filtering duplicates and linear combinations significantly reduces the number $c$ of candidates, we usually still have to select a subset of $b < c$. Thus, we have $\binom{c}{b}$ possible $b$ sized subsets, each inducing a candidate $Q'$ and thus $X'$. A naive approach is to compute the gradients for all $X'$ and compare them to the ground truth. However, this is computationally infeasible even for moderate $c$.

To address this, we propose a greedy two-stage procedure optimizing a novel sparsity matching score $\lambda$, which resolves the computational complexity issue above while also accurately selecting the correct batch elements and relying solely on $\frac{\partial \mathcal{L}}{\partial Z}'$ and $Z'$. As both can be computed directly via $Q'$, the procedure is local and does not need to backpropagate gradients. Next, we explain the first stage.

**Dictionary Learning [10]** As a first stage, we leverage a component of the algorithm proposed by Spielman et al. [10] for sparsely-used dictionary learning. This approach is based on the insight that the subset of column vectors $\mathcal{B} = \{\overline{q}'_i\}_{i=1}^b$, yielding the sparsest full-rank gradient matrix $\frac{\partial L}{\partial Z}$ is often correct. As the scaling of $\overline{q}'_i$ does not change the sparsity of the resulting $\frac{\partial L}{\partial Z}$, we can construct the subset $\mathcal{B}$ by greedily collecting the $b$ directions $\overline{q}'_i$ with the highest corresponding sparsity that still increase the rank of $\mathcal{B}$. While this method typically recovers most directions $\overline{q}_i$, it often misses directions whose gradients $\frac{\partial L}{\partial z_i}$ are less sparse by chance.

**Sparsity Matching** We alleviate this issue by introducing a second stage to the algorithm where we greedily optimize a novel correctness measure based solely on the gradients of the linear layer, which we call the sparsity matching coefficient $\lambda$.

**Definition 4.1.** Let $\lambda_-$ be the number of non-positive entries in $Z$ whose corresponding entries in $\frac{\partial \mathcal{L}}{\partial Z}$ are $0$. Similarly, let $\lambda_+$ be the number of positive entries in $Z$ whose corresponding entries in $\frac{\partial \mathcal{L}}{\partial Z}$ are not $0$. We call their normalized sum the *sparsity matching coefficient* $\lambda$:

$$\lambda = \frac{\lambda_- + \lambda_+}{m \cdot b}.$$

Intuitively, this describes how well the pre-activation values $Z$ match the sparsity pattern of the gradients $\frac{\partial \mathcal{L}}{\partial Z}$ induced by the ReLU layer (See Sec. 3.2). While this sparsity matching coefficient $\lambda$ can take values between 0 and 1, it is exactly $\lambda = 1$ for the correct $X$, if the gradient $\frac{\partial \mathcal{L}}{\partial Y}$ w.r.t. the ReLU output is dense, which is usually the case. We note that $\lambda$ can be computed efficiently for arbitrary full rank matrix $\overline{Q}'$ by computing $\frac{\partial \mathcal{L}}{\partial Z}' = LQ'$ and $Z' = WX' + (b|\dots|b)$ for $X'^\top = Q'^{-1}R$.

To optimize $\lambda$, we initialize $\overline{Q}'$ with the result of the greedy algorithm in Spielman et al. [10], and then greedily swap the pair of vectors $\overline{q}'_i$ improving $\lambda$ the most, while keeping the rank, until convergence.

# 5 Final Algorithm and Complexity Analysis

In this section, we first present our final algorithm SPEAR (Sec. 5.1) and then analyse its expected complexity and failure probability (Sec. 5.2).

## 5.1 Final Algorithm

We formalize our gradient inversion attack SPEAR in Alg. 1 and outline it below. First, we compute the low-rank decomposition $\frac{\partial \mathcal{L}}{\partial \boldsymbol{W}} = \boldsymbol{L}\boldsymbol{R}$ of the weight gradient $\frac{\partial \mathcal{L}}{\partial \boldsymbol{W}}$ via reduced SVD, allowing us to recover the batch size $b$ as the rank of $\frac{\partial \mathcal{L}}{\partial \boldsymbol{W}}$ (Line 2). We now sample (at most $N$) submatrices $\boldsymbol{L_A}$ of $\boldsymbol{L}$ and compute proposal directions $\bar{\boldsymbol{q}}'_i$ as their kernel $\ker(\boldsymbol{L_A})$ via SVD (Lines 4–5). We note that our implementation parallelizes both sampling and SVD computation (Lines 4–5) on a GPU. We then filter the proposal directions $\bar{\boldsymbol{q}}'_i$ based on their sparsity (Line 6), adding them to our candidate pool $\mathcal{C}$

**Algorithm 1** SPEAR

1: **function** SPEAR( m, n, $\boldsymbol{W}$, $\boldsymbol{b}$, $\frac{\partial \mathcal{L}}{\partial \boldsymbol{W}}$, $\frac{\partial \mathcal{L}}{\partial \boldsymbol{b}}$ )
2:     $\boldsymbol{L}, \boldsymbol{R}, b \leftarrow$ LOWRANKDECOMPOSE $\left(\frac{\partial \mathcal{L}}{\partial \boldsymbol{W}}\right)$
3:     **for** $i = 1$ **to** $N$ **do**
4:         Sample a submatrix $\boldsymbol{L_A} \in \mathbb{R}^{b-1 \times b}$ of $\boldsymbol{L}$
5:         $\bar{\boldsymbol{q}}'_i \leftarrow \ker(\boldsymbol{L_A})$
6:         **if** sparsity$(\boldsymbol{L}\bar{\boldsymbol{q}}'_i) \geq \tau * m$ **and** $\bar{\boldsymbol{q}}'_i \notin \mathcal{C}$ **then**
7:             $\mathcal{C} \leftarrow \mathcal{C} \cup \{\bar{\boldsymbol{q}}'_i\}$
8:             $\lambda, \boldsymbol{X}' \leftarrow$ GREEDYFILT $(\boldsymbol{L}, \boldsymbol{R}, \boldsymbol{W}, \boldsymbol{b}, \frac{\partial \mathcal{L}}{\partial \boldsymbol{b}}, \mathcal{C})$
9:             **if** $\lambda = 1$ **then**
10:                **return** $\boldsymbol{X}'$
11:             **end if**
12:         **end if**
13:     **end for**
14:     $\lambda, \boldsymbol{X}' \leftarrow$ GREEDYFILT $(\mathcal{C})$
15:     **return** $\boldsymbol{X}'$

if they haven't been recovered already and are sufficiently sparse (Line 7). Once our candidate pool contains at least $b$ directions, we begin constructing candidate input reconstructions $\boldsymbol{X}'$ using our two-stage greedy algorithm GREEDYFILTER (Line 8), discussed in Sec. 4.2. If this reconstruction leads to a solution with sparsity matching coefficient $\lambda = 1$, we terminate early and return the corresponding solution (Line 9). Otherwise, we continue sampling until we have reached $N$ samples and return the best reconstruction we can obtain from the resulting candidate pool (Line 14). The pseudocode for COMPUTESIGMA (Alg. 2) and GREEDYFILTER (Alg. 3) are shown in App. C.

## 5.2 Analysis

In this section, we will analyze SPEAR w.r.t. the number of submatrices we *expect* to sample until we have recovered all $b$ correct directions $\bar{\boldsymbol{q}}_i$ (Lemma 5.2), and the probability of failing to recover all $b$ correct directions despite checking all possible submatrices of $\boldsymbol{L}$ (Lemma 5.3). For an analysis of the number of submatrices we have to sample until we have recovered all $b$ correct directions $\bar{\boldsymbol{q}}_i$ *with high probability*, we point to Lemma B.2. Further, as before, we defer all proofs also to App. B.

**Expected Number of Required Samples**
To determine the expected number of required samples until we have recovered the correct $b$ direction vectors $\bar{\boldsymbol{q}}_i$, we first compute a lower bound on the probability $q$ of sampling a submatrix which satisfies the conditions of Theorem 3.3 for an arbitrary column $i$ in $\overline{\boldsymbol{Q}}$ and then use the coupon collector problem to compute the expected number of required samples.

We can lower bound the probability of a submatrix $\boldsymbol{A} \in \mathbb{R}^{b-1 \times b}$, randomly sampled as $b-1$ rows of $\frac{\partial \mathcal{L}}{\partial \boldsymbol{Z}}$, having exactly one all-zero column and being full rank as follows:

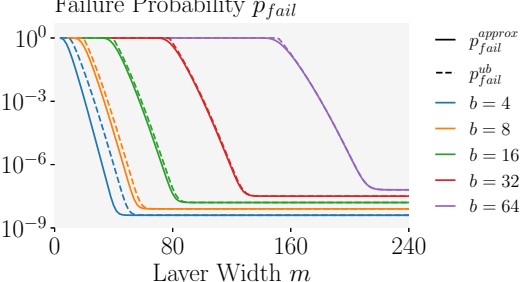

Failure Probability $p_{fail}$

Figure 3: Visualizations of the upper bound ($p_{\text{fail}}^{\text{ub}}$, dashed) on and approximation of ($p_{\text{fail}}^{\text{approx}}$, solid) the failure probability of SPEAR for different batch sizes $b$ and network widths $m$ for $p_{fr} = 10^{-9}$.

**Lemma 5.1.** *Let* $\boldsymbol{A} \in \mathbb{R}^{b-1 \times b}$ *be submatrix of the gradient* $\frac{\partial \mathcal{L}}{\partial \boldsymbol{Z}}$ *obtained by sampling* $b-1$ *rows uniformly at random without replacement, where each element of* $\frac{\partial \mathcal{L}}{\partial \boldsymbol{Z}}$ *is distributed i.i.d. as* $\frac{\partial \mathcal{L}}{\partial \boldsymbol{Z}_{j,k}} = \zeta|\epsilon|$ *with* $\epsilon \sim \mathcal{N}(\mu = 0, \sigma^2 > 0)$ *and* $\zeta \sim Bernoulli(p = \frac{1}{2})$. *We then have the probability*

*q* of $\boldsymbol{A}$ *having exactly one all-zero column and being full rank lower bounded by:*

$$q \geq \frac{b}{2^{b-1}} \left(1 - (\tfrac{1}{2} + o_{b-1}(1))^{b-1}\right) \geq \frac{b}{2^{b-1}}(1 - 0.939^{b-1}).$$

We can now compute the expected number of submatrices $n_{\text{total}}^*$ we have to draw until we have recovered all $b$ correct direction vectors using the Coupon Collector Problem:

**Lemma 5.2.** *Assuming i.i.d. submatrices $\boldsymbol{A}$ following the distribution outlined in Lemma 5.1 and using Alg. 1, we have the expected number of submatrices $n_{total}^*$ required to recover all $b$ correct direction vectors as:*

$$n_{total}^* = \frac{1}{q} \sum_{k=0}^{b-1} \frac{b}{b-k} = \frac{bH_b}{q} \approx \frac{1}{q}(b\log(b) + \gamma b + \tfrac{1}{2}),$$

*where $H_b$ is the $b^{th}$ harmonic number and $\gamma \approx 0.57722$ the Euler-Mascheroni constant.*

We validate this result experimentally in Fig. 4 where we observe excellent agreement for wide networks ($m \gg b$) and obtain, e.g., $n_{\text{total}}^* \approx 1.8 \times 10^5$ for a batch size of $b = 16$.

**Failure Probability** We now analyze the probability of SPEAR failing despite considering all possible submatrices of $\boldsymbol{L}$ and obtain:

**Lemma 5.3.** *Under the same assumptions as in Lemma 5.1, we have an upper bound on the failure probability $p_{fail}^{ub}$ of Alg. 1 even when sampling exhaustively as:*

$$p_{fail}^{ub} \leq b \left(1 - \sum_{k=b-1}^{m} \binom{m}{k} \frac{1}{2^m} \left(1 - 0.939^{(b-1)\binom{k}{b-1}}\right)\right) + 1 - (1 - p_{fr})^b,$$

*where $p_{fr}$ is the probability of falsely rejecting a correct direction $\bar{\boldsymbol{q}}'$ via our sparsity filter (Sec. 4.1).*

If we assume the full-rankness of submatrices $\boldsymbol{A}$ to i) occur with probability $1 - (\tfrac{1}{2} - o_{b-1}(1))^{b-1}$ for $o_{b-1}(1) \approx 0$ (true for large $b$ [11]) and ii) be independent between submatrices, we instead obtain:

$$p_{fail}^{approx} \approx 1 - \left(\sum_{k=b-1}^{m} \binom{m}{k} \frac{1}{2^m} \left(1 - 0.5^{(b-1)\binom{k}{b-1}}\right)\right)^b + 1 - (1 - p_{fr})^b.$$

We illustrate this bound in Fig. 3 and empirically validate this bound in Fig. 8 and observe the true failure probability to lie between $p_{fail}^{approx}$ and $p_{fail}^{ub}$.

# 6 Empirical Evaluation

In this section, we empirically evaluate the effectiveness of SPEAR on MNIST [13], CIFAR-10 [14], TINYIMA-GENET [15], and IMAGENET [16] across a wide range of settings. In addition to the reconstruction quality metrics PSNR and LPIPS, commonly used to evaluate gradient inversion attacks, we report accuracy as the portion of batches for which we recovered the batch up to numerical errors and the number of sampled submatrices (number of iterations).

Table 1: Comparison to prior work in the image domain.

| Method | PSNR ↑ | Time/Batch |
|---|---|---|
| CI-Net [12] Sigmoid | 38.0 | 1.6 hrs |
| CI-Net [12] ReLU | 15.6 | 1.6 hrs |
| Geiping et al. [1] | 19.6 | 18.0 min |
| SPEAR (Ours) | **124.2** | **2.0 min** |

**Experimental Setup** For all experiments, we use our highly parallelized PyTorch [17] GPU implementation of SPEAR. Unless stated otherwise, we run all experiments on CIFAR-10 batches of size $b = 20$ using a 6 layer ReLU-activated FCNN with width $m = 200$ and set $\tau$ to achieve a false rejection rate of $p_{fr} \leq 10^{-5}$. We supply ground truth labels to all methods *except* SPEAR.

## 6.1 Comparison to Prior Work

In Table 1, we compare SPEAR against prior gradient inversion attacks from the image domain on the IMAGENET dataset rescaled to $256 \times 256$ resolution. In particular, we compare to Geiping et al. [1][1], as well as, the recent CI-Net [12]. As CI-Net only considers networks with the less common Sigmoid activations, we report its performance on both ReLU and Sigmoid versions of our network.

---

[1]We use so-called "modern" version of the attack from https://github.com/JonasGeiping/breaching

We observe that while CI-Net obtains very good reconstructions with the Sigmoid network (PSNR of 38), SPEAR still achieves a much higher PSNR (124) as it is exact. Further, for the more common ReLU acti-

Table 2: Results vs prior work in the tabular domain.

| Method | Discr Acc (%) ↑ | Cont. MAE ↓ | Time/Batch |
|---|---|---|---|
| Tableak [8] | 97 | 4922.7 | 2.6 min |
| SPEAR (Ours) | **100** | **20.4** | **0.4 min** |

vations, the performance of CI-Net drops significantly to a PSNR $< 16$ compared to 19.6 for Geiping et al. [1]. Additionally, SPEAR is much faster compared to both Geiping et al. [1] and CI-Net, taking $10\times$ and $100\times$ less time, respectively. Finally, we want to emphasize that both prior works rely on strong prior knowledge, including label information and knowledge of the structure of images, whereas we assume *no information at all* about the data distribution and still achieve much better results in only a fraction of the time taken.

To confirm the versatility of SPEAR, we compare it to the SoTA attack in the tabular domain, Tableak [8], in Table 2. We see that due to the exact nature of our attack, we recover both continuos and discrete features better on the ADULT dataset [18] with $b = 16$, while still being $6\times$ faster.

## 6.2 Main Results

We evaluate SPEAR on MNIST, CIFAR-10, TINYIMAGENET and IMAGENET at two different res-

Table 3: Reconstruction quality across 100 batches.

| Dataset | PSNR ↑ | LPIPS ↓ | Acc (%) ↑ | Time/Batch |
|---|---|---|---|---|
| MNIST | 99.1 | NaN | 99 | 2.6 min |
| CIFAR-10 | 106.6 | $1.16 \times 10^{-5}$ | 99 | 1.7 min |
| TINYIMAGENET | 110.7 | $1.62 \times 10^{-4}$ | 99 | **1.4 min** |
| IMAGENET $224 \times 224$ | 125.4 | $1.05 \times 10^{-5}$ | 99 | 2.1 min |
| IMAGENET $720 \times 720$ | **125.6** | $8.08 \times 10^{-11}$ | 99 | 2.6 min |

olutions, reporting results in Table 3. Across datasets, SPEAR can reconstruct almost all batches perfectly, achieving PSNRs of 100 and above even at a batch size of $b = 20$ for images as large as $720 \times 720$ in $< 3$ minutes. We provide additional results on heterogeneous data and trained networks in App. E, as well as, on the FedAvg protocol in App. F.

**Effect of Batch Size $b$**  We evaluate the effect of batch size $b$ on accuracy and the required number of iterations $n^*_{total}$ for a wide ($m = 2000$) and narrow ($m = 200$) network. While $n^*_{total}$ increases exponentially with $b$, for both networks, the narrower network requires about 20 times more iterations than the wider network (see Fig. 4). While trends for the wider network ($m \gg b$) are perfectly described by our theoretical results in Sec. 5.2, some independence assumptions are violated for the narrower network, explaining the larger number of required iterations. While we can recover all batches perfectly for the wider network, we see a sharp drop in accuracy from 99% at $b = 20$ to 63% at $b = 24$ (see Fig. 6) for the narrower network. This is due to increasingly more batches requiring more than the $N = 2 \times 10^9$ submatrices we sample at most.

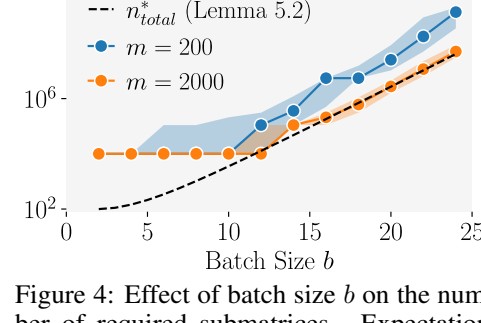

Figure 4: Effect of batch size $b$ on the number of required submatrices. Expectation from Lemma 5.2 dashed and median ($10^{th}$ to $90^{th}$ percentile shaded) depending on network width $m$ solid. We always evaluate $10^4$ submatrices in parallel, explaining the plateau.

**Effect of Network Architecture**  We visualize the performance of SPEAR across different network widths and depths in Fig. 5. We observe that while accuracy is independent of both (given sufficient width $m \gg b$), the number of required iterations reduces with increasing width $m$. We provide further ablations on the effect of our two-stage filtering in App. E.3 and DPSGD noise in App. E.6.

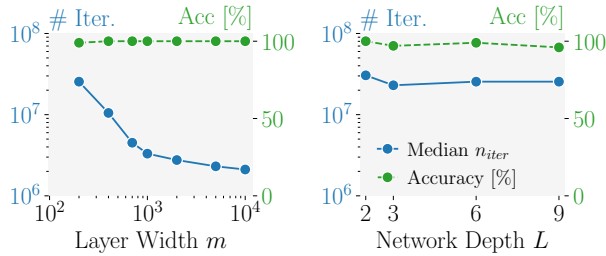

Figure 5: Accuracy (green) and number of median iterations (blue) for different network widths $m$ at $L = 6$ (left) and depths $L$ at $m = 200$ (right).

**Effect of Layer Depth**  Our experiments so far focused on recovering inputs to the first layer of FCNNs. However, SPEAR's capabilities extend beyond this, as highlighted in Sec. 3.2. To

demonstrate this, we use SPEAR to reconstruct the inputs to all FC layers followed by a ReLU activation in a 6-layer FCNN with a width of $m = 400$ at initialization.

The results, presented in Table 4, show that SPEAR successfully recovers the inputs to all layers almost perfectly. However, attacking later layers is more computationally expensive. Specifically, the runtime for $l = 5$ increases to 70 minutes/batch resulting in 17 batches that timed-out. This increased computational cost is due to the initialization of the network, which causes the outputs of later layers to be dominated by their bias terms with their inputs being almost irrelevant. This issue is mitigated after a few training steps, as weights and biases adjust to better reflect the relationships between inputs and outputs. We find that after 5000 gradient steps the time per batch reduces to $< 1$ min at an accuracy of $> 95\%$ for layer $l = 5$.

Table 4: Effect of the attacked layer's depth $l$ ($1 \leq l \leq 6$) on reconstruction time and quality for 100 TINYIMAGENET batches of size $b = 20$.

| $l$ | MAE $\downarrow$ | Acc(%) $\uparrow$ | Time/Batch |
|---|---|---|---|
| 1 | $\mathbf{1.06 \times 10^{-6}}$ | **100** | 2.3 min |
| 2 | $1.33 \times 10^{-6}$ | **100** | **2.2 min** |
| 3 | $1.67 \times 10^{-6}$ | **100** | 5.6 min |
| 4 | $2.80 \times 10^{-6}$ | 99 | 19 min |
| 5 | $3.04 \times 10^{-6}$ | 83 | 70 min |

## 6.3 Scaling SPEAR via Optimization-based Attacks

As we prove theoretically in Sec. 5.2 and verify practically in App. E.5, in the common regime where the batch size $b$ is much smaller than dimensions of the attacked linear layer w.h.p. the input information is losslessly represented in the client gradient. However, in practice for $b > 25$ the exponential sampling complexity of SPEAR becomes a bottleneck that prevents the recovery of the input (see Fig. 4).

Table 5: Comparison between the reconstruction quality of Geiping et al. [1] and a version of SPEAR that uses Geiping et al. [1] to speed up its search procedure evaluated on 10 TINYIMAGENET batches.

| Method | $b$ | $m$ | Acc(%) $\uparrow$ | PSNR $\uparrow$ |
|---|---|---|---|---|
| Geiping et al. [1] | 50 | 400 | - | 26.5 |
| SPEAR + Geiping et al. [1] | 50 | 400 | 100 | **124.5** |
| Geiping et al. [1] | 100 | 2000 | - | 32.8 |
| SPEAR + Geiping et al. [1] | 100 | 2000 | 60 | **81.5** |

In this section, we propose a method for alleviating the exponential sampling complexity by combining SPEAR with an approximate reconstruction method to get a prior on which submatrices $\boldsymbol{L}_A$ satisfy the conditions of Theorem 3.3, i.e., have corresponding matrices $\boldsymbol{A}$ containing a 0-column. To this end, we first obtain an estimate of the client pre-activation values $\widetilde{\boldsymbol{Z}}$ based on the approximate input reconstructions from Geiping et al. [1]. As large negative pre-activation values in $\widetilde{\boldsymbol{Z}}$ are much more likely to correspond to negative pre-activation values in the true $\boldsymbol{Z}$, and, thus, to 0s in gradients $\frac{\partial \mathcal{L}}{\partial \boldsymbol{Z}}$, we record the locations of the $3b$ largest negative values for each column of $\widetilde{\boldsymbol{Z}}$. Importantly, by choosing the locations this way, we ensure that each group of $3b$ locations correspond to locations of likely 0s in *same column* of $\frac{\partial \mathcal{L}}{\partial \boldsymbol{Z}}$. Restricting the sampling of the row indices of $\boldsymbol{L}_A$ and $\boldsymbol{A}$ only within each group of locations, ensures that $\boldsymbol{L}_A$ is very likely to satisfying the conditions of Theorem 3.3.

We confirm the effectiveness of this approach in a preliminary study, shown in Table 5, that demonstrates the combined approach allows a substantial increase in the batch size SPEAR can scale to (up to 100), thus effectively eliminating its exponential complexity. The results show that the combined approach drastically improves the reconstruction quality of Geiping et al. [1] as well, as unlike Geiping et al. [1], it achieves exact reconstruction. Importantly, we observe that even for the 4 batches SPEAR failed to recover in Table 5, SPEAR still reconstructs $> 97$ of the 100 directions $\overline{\boldsymbol{q}_i}$ correctly, suggesting that future work can further improve upon our results.

## 6.4 Feature Inversion in Convolutional Neural Networks

Following the Cocktail Party Attack (CPA) [9], we experiment with using SPEAR to recover the input features to the first linear layer of a pretrained VGG16 convolutional network with size $25088 \times 4096$ for IMAGENET batches of $b = 16$ and use them in a feature inversion (FI) attack to approximately recover the client images. We

Table 6: Comparison between the reconstructions on VGG16 for Geiping et al. [1], CPA [9], and SPEAR for 10 IMAGENET batches ($b = 16$).

| Method | LPIPS $\downarrow$ | Feature Sim $\uparrow$ |
|---|---|---|
| Geiping et al. [1] | 0.562 | - |
| CPA[9] + FI + Geiping et al. [1] | 0.388 | 0.939 |
| SPEAR + FI + Geiping et al. [1] | **0.362** | **0.984** |

show the results of our experiments, based on the CPA's code and parameters, in Table 6. We see the inverted features drastically improve quality of the final reconstructions, and that SPEAR achieves almost perfect feature cosine similarity, resulting in better overall reconstruction versus CPA.

# 7 Related Work

In this section, we discuss how we relate to prior work.

**Gradient Inversion Attacks** Since gradient inversion attacks have been introduced [3], two settings have emerged: In the *malicious setting*, the server does not adhere to the training protocol and can adversarially engineer network weights that maximize leaked information [19, 20, 21, 22]. In the strictly harder *honest-but-curious setting*, the server follows the training protocol but still aims to reconstruct client data. We target the honest-but-curious setting, where prior work has either recovered the input exactly for batch sizes of $b = 1$ [5, 6], or approximately for $b > 1$ [1, 23, 7, 8, 9]. In this setting, we are the *first to reconstruct inputs exactly for batch sizes $b > 1$*.

Most closely related to our work is Kariyappa et al. [9] which leverage the low-rank structure of the gradients to frame gradient inversion as a blind source separation problem, improving their approximate reconstructions. In contrast, we derive an *explicit* low-rank representation and additionally leverage gradient sparsity reconstruct inputs exactly.

Unlike a long line of prior work, we rely neither on any priors on the data distribution [8, 24, 25, 26] nor on a reconstructed classification label [1, 27, 7, 23, 8]. This allows our approach to be employed in a much wider range of settings where neither is available.

**Defenses Against Gradient Inversion** Defenses based on Differential Privacy [28] add noise to the computed gradients on the client side, providing provable privacy guarantees at the cost of significantly reduced utility. Another line of work increases the empirical difficulty of inversion by increasing the effective batch size, by securely aggregating gradients from multiple clients [29] or doing multiple gradient update steps locally before sharing an aggregated weight update [4]. Finally, different heuristic defenses such as gradient pruning [3] have been proposed, although their effectiveness has been questioned [30].

**Sparsely-used Dictionary learning** Recovering the disaggregation matrix $Q$ is related to the well-studied problem of sparsely-used dictionary learning. However, there the aim is to find the sparsest coefficient matrix (corresponding to our $\frac{\partial \mathcal{L}}{\partial Z}$) and dense dictionary ($Q^{-1}$) approximately encoding a signal ($L$). In contrast, we do not search for the sparsest solution yielding an approximate reconstruction but a solution that exactly induces consistent $X$ and $\frac{\partial \mathcal{L}}{\partial Z}$, which happens to be sparse. Sparsely-used dictionary learning is known to be NP-hard [31] and typically solved approximately [32, 10, 33]. However, under sufficient sparsity, it can be solved exactly in polynomial time [10]. While our $\frac{\partial \mathcal{L}}{\partial Z}$ are not sparse enough, we still draw inspiration from Spielman et al. [10] in Sec. 4.

# 8 Limitations

We focus on recovering the inputs to fully connected layers with ReLU activations such as they occur at the beginning of fully connected networks or as aggregation layers of many other architectures. Extending our approach to other layers is an interesting direction for future work.

Further, our approach scales exponentially with batch size $b$. While SPEAR's massive parallelizability and its ability to be combined with optimization-based attacks, as shown in Sec. 6.3, can partially mitigate the computational complexity, future research is still required to make reconstruction of batches of size $b > 100$ practical.

# 9 Conclusion

We propose SPEAR, the first algorithm permitting batches of $b > 1$ elements to be recovered exactly in the honest-but-curious setting. We demonstrate theoretically and empirically that SPEAR succeeds with high probability and that our highly parallelized GPU implementation is effective across a wide range of settings, including batches of up to 25 elements and large networks and inputs.

We thereby demonstrate that contrary to prior belief, an exact reconstruction of batches is possible in the honest-but-curious setting, suggesting that federated learning on ReLU networks might be inherently more susceptible than previously thought. To still protect client privacy, large effective batch sizes, obtained, e.g., via secure aggregation across a large number of clients, might prove instrumental by making reconstruction computationally intractable.

**Acknowledgments**

This research was partially funded by the Ministry of Education and Science of Bulgaria (support for INSAIT, part of the Bulgarian National Roadmap for Research Infrastructure).

This work has been done as part of the EU grant ELSA (European Lighthouse on Secure and Safe AI, grant agreement no. 101070617) . Views and opinions expressed are however those of the authors only and do not necessarily reflect those of the European Union or European Commission. Neither the European Union nor the European Commission can be held responsible for them.

The work has received funding from the Swiss State Secretariat for Education, Research and Innovation (SERI).

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

## A  Broader Impact

In this work, we demonstrate that contrary to prior belief, an exact reconstruction of batches is possible in the honest-but-curious setting for federated learning. As our work demonstrates the susceptibility of federated learning systems using ReLU networks, this work inevitably advances the capabilities of an adversary. Nonetheless, we believe this to be an important step in accurately assessing the risks and utilities of federated learning systems.

To still protect client privacy, large effective batch sizes, obtained, e.g., via secure aggregation across a large number of clients, might prove instrumental by making reconstruction computationally intractable. As gradient information and network states can be stored practically indefinitely, our work highlights the importance of proactively protecting client privacy in federated learning not only against current but future attacks. This underlines the importance of related work on provable privacy guarantees obtained via differential privacy.

## B  Deferred Proofs

**Theorem 3.1.** *The network's gradient w.r.t. the weights $\boldsymbol{W}$ can be represented as the matrix product:*

$$\frac{\partial \mathcal{L}}{\partial \boldsymbol{W}} = \frac{\partial \mathcal{L}}{\partial \boldsymbol{Z}} \boldsymbol{X}^T. \tag{1}$$

*Proof.* We will use Einstein notation for this proof:

$$
\begin{aligned}
\frac{\partial \mathcal{L}}{\partial \boldsymbol{W}_i{}^j} &= \frac{\partial \mathcal{L}}{\partial \boldsymbol{Z}_k{}^l} \frac{\partial \boldsymbol{Z}_k{}^l}{\partial \boldsymbol{W}_i{}^j} \\
&= \frac{\partial \mathcal{L}}{\partial \boldsymbol{Z}_k{}^l} \frac{\partial (\boldsymbol{W}_k{}^m \boldsymbol{X}_m{}^l + b_k \delta^l)}{\partial \boldsymbol{W}_i{}^j} \\
&= \frac{\partial \mathcal{L}}{\partial \boldsymbol{Z}_k{}^l} \frac{\partial \boldsymbol{W}_k{}^m \boldsymbol{X}_m{}^l}{\partial \boldsymbol{W}_i{}^j} \\
&= \frac{\partial \mathcal{L}}{\partial \boldsymbol{Z}_k{}^l} \frac{\partial \boldsymbol{W}_k{}^m}{\partial \boldsymbol{W}_i{}^j} \boldsymbol{X}_m{}^l \\
&= \frac{\partial \mathcal{L}}{\partial \boldsymbol{Z}_k{}^l} \delta_k{}^i \delta_j{}^m \boldsymbol{X}_m{}^l \\
&= \frac{\partial \mathcal{L}}{\partial \boldsymbol{Z}_i{}^l} \boldsymbol{X}_j{}^l \\
&= \frac{\partial \mathcal{L}}{\partial \boldsymbol{Z}_i{}^l} (\boldsymbol{X}^T)^l{}_j.
\end{aligned}
$$

We note that $\delta_k{}^i$ is the Kronecker delta, that is $\delta_k{}^i = 1$ if $k = i$ and 0 otherwise. Further, $\delta^l = 1$ for all $l$. Hence we arrive at Eq. 1. $\qquad\square$

**Lemma B.1.** *Let $b, n, m \in \mathbb{N}$ such that $b < n, m$. Further, let $\boldsymbol{A}, \boldsymbol{L} \in \mathbb{R}^{m \times b}$ and $\boldsymbol{B}, \boldsymbol{R} \in \mathbb{R}^{b \times n}$ be matrices of maximal rank, satisfying $\boldsymbol{AB} = \boldsymbol{LR}$. Then there exists a unique disaggregation matrix $\boldsymbol{Q} \in GL_b(\mathbb{R})$ s.t. $\boldsymbol{A} = \boldsymbol{LQ}$, and $\boldsymbol{B} = \boldsymbol{Q}^{-1} \boldsymbol{R}$.*

*Proof.* As $b \leq n, m$ and the matrices $\boldsymbol{A} \in \mathbb{R}^{m \times b}$ and $\boldsymbol{B} \in \mathbb{R}^{b \times n}$ have full rank, we know that there exists

- a left inverse $\boldsymbol{A}^{-L} \in \mathbb{R}^{b \times m}$ for $\boldsymbol{A}$: $\boldsymbol{A}^{-L} \boldsymbol{A} = \boldsymbol{I}_b$ and
- a right inverse $\boldsymbol{B}^{-R} \in \mathbb{R}^{n \times b}$ for $\boldsymbol{B}$: $\boldsymbol{B} \boldsymbol{B}^{-R} = \boldsymbol{I}_b$.

Thus, it follows from

$$\boldsymbol{A}^{-L} \boldsymbol{L} \boldsymbol{R} \boldsymbol{B}^{-R} = \boldsymbol{A}^{-L} \boldsymbol{A} \boldsymbol{B} \boldsymbol{B}^{-R} = \boldsymbol{I}_b,$$

that $(\boldsymbol{A}^{-L} \boldsymbol{L})^{-1} = \boldsymbol{R} \boldsymbol{B}^{-R}$. We now set $\boldsymbol{Q} = \boldsymbol{R} \boldsymbol{B}^{-R}$.

This $Q$ satisfies the required properties:

- $B = Q^{-1}R$:
$$Q^{-1}R = A^{-L}LR = A^{-L}AB = B,$$

- $A = LQ$:
$$LQ = LRB^{-R} = ABB^{-R} = A,$$

- Uniqueness: Assume we have $Q_1$ and $Q_2$ that satisfy $LQ_1 = LQ_2 = A$. As $L$ is of rank $b$ and $b \le m$, there exists a left inverse $L^{-L}$ for $L$: $L^{-L}L = I_b$. Applying this left inverse to $LQ_1 = LQ_2$, directly yields $Q_1 = Q_2$, and hence we get uniqueness.

$\square$

**Theorem 3.4.** *The gradient w.r.t. the bias $b$ can be written in the form $\frac{\partial \mathcal{L}}{\partial b} = \frac{\partial \mathcal{L}}{\partial Z}\begin{bmatrix} 1 \\ \vdots \\ 1 \end{bmatrix}$.*

*Proof.* We use again Einstein notation.

$$
\begin{aligned}
\frac{\partial \mathcal{L}}{\partial b_i} &= \frac{\partial \mathcal{L}}{\partial Z_k{}^l} \frac{\partial Z_k{}^l}{\partial b_i} \\
&= \frac{\partial \mathcal{L}}{\partial Z_k{}^l} \frac{\partial (W_k{}^m X_m{}^l + b_k \delta^l)}{\partial b_i} \\
&= \frac{\partial \mathcal{L}}{\partial Z_k{}^l} \frac{\partial b_k \delta^l}{\partial b_i} \\
&= \frac{\partial \mathcal{L}}{\partial Z_k{}^l} \delta_k{}^i \delta^l \\
&= \frac{\partial \mathcal{L}}{\partial Z_i{}^l} \delta^l.
\end{aligned}
$$

This concludes the proof. $\square$

**Theorem 3.5.** *For any left inverse $L^{-L}$ of $L$, we have $\begin{bmatrix} s_1 \\ \vdots \\ s_b \end{bmatrix} = \overline{Q}^{-1}L^{-L}\frac{\partial \mathcal{L}}{\partial b}$*

*Proof.* The proof is straight forward. Using Theorem 3.4 and Theorem 3.2, we know that

$$
\begin{aligned}
\overline{Q}^{-1}L^{-L}\frac{\partial \mathcal{L}}{\partial b} &= \overline{Q}^{-1}L^{-L}\frac{\partial \mathcal{L}}{\partial Z}\begin{bmatrix} 1 \\ \vdots \\ 1 \end{bmatrix} \\
&= \overline{Q}^{-1}L^{-L}LQ\begin{bmatrix} 1 \\ \vdots \\ 1 \end{bmatrix} \\
&= \overline{Q}^{-1}Q\begin{bmatrix} 1 \\ \vdots \\ 1 \end{bmatrix} \\
&= \overline{Q}^{-1}\overline{Q}\operatorname{diag}(s_1,\ldots,s_b)\begin{bmatrix} 1 \\ \vdots \\ 1 \end{bmatrix} \\
&= \begin{bmatrix} s_1 \\ \vdots \\ s_b \end{bmatrix}.
\end{aligned}
$$

$\square$

**Lemma 5.1.** *Let $A \in \mathbb{R}^{b-1 \times b}$ be submatrix of the gradient $\frac{\partial \mathcal{L}}{\partial Z}$ obtained by sampling $b-1$ rows uniformly at random without replacement, where each element of $\frac{\partial \mathcal{L}}{\partial Z}$ is distributed i.i.d. as $\frac{\partial \mathcal{L}}{\partial Z_{j,k}} = \zeta|\epsilon|$ with $\epsilon \sim \mathcal{N}(\mu = 0, \sigma^2 > 0)$ and $\zeta \sim$ Bernoulli$(p = \frac{1}{2})$. We then have the probability $q$ of $A$ having exactly one all-zero column and being full rank lower bounded by:*

$$q \ge \frac{b}{2^{b-1}}\left(1 - (\tfrac{1}{2} + o_{b-1}(1))^{b-1}\right) \ge \frac{b}{2^{b-1}}(1 - 0.939^{b-1}).$$

*Proof.* We have the probability of one of the $b$ columns being all zero as $\frac{b}{2^{b-1}}$ if the network has full rank, all other columns will not be all-zero.

Further, we have the probability of the submatrix $\mathbb{1}_{\boldsymbol{A}>0}$ being full rank conditioned on column $i$ being all-zero as the probability of the matrix described by remaining $b-1$ columns being non-singular. This probability is $1 - (\frac{1}{2} + o_{b-1}(1))^{b-1}$ [11] where $\lim_{b\to\infty} o_{b-1}(1) = 0$, which can be lower-bounded with $1 - 0.939^{b-1}$ [34]. We thus obtain their joint probability as their product. $\qquad\square$

**Lemma 5.2.** *Assuming i.i.d. submatrices $\boldsymbol{A}$ following the distribution outlined in Lemma 5.1 and using Alg. 1, we have the expected number of submatrices $n^*_{total}$ required to recover all $b$ correct direction vectors as:*

$$n^*_{total} = \frac{1}{q} \sum_{k=0}^{b-1} \frac{b}{b-k} = \frac{bH_b}{q} \approx \frac{1}{q}(b\log(b) + \gamma b + \tfrac{1}{2}),$$

*where $H_b$ is the $b^{th}$ harmonic number and $\gamma \approx 0.57722$ the Euler-Mascheroni constant.*

*Proof.* As we sample submatrices $\boldsymbol{A}$ uniformly at random with replacement, assuming them to be i.i.d. is well justified for the regime of $m \gg b$. The the number $n$ of submatrices drawn between correct direction vectors $\boldsymbol{q}_i$ thus follows a Geometric distribution $\mathbb{P}[n = k] = q(1-q)^{k-1}$ with success probability $q$ with expectation $n^* = \mathbb{E}[n] = \frac{1}{q}$. As we draw correct direction vectors $\boldsymbol{q}_i$ uniformly at random from the $b$ columns of $\overline{\boldsymbol{Q}}$, we have the probability of drawing a new direction vector $\boldsymbol{q}_i$ as $\frac{b-k}{b}$ for $k$ already drawn direction vectors. Again via the expectation of the Geometric distribution we obtain the expected number $c^*$ of correct direction vectors we have to draw until we have recovered all $b$ distinct ones as the solution of the Coupon Collector Problem $c^* = \sum_{k=0}^{b-1} \frac{b}{b-k} = bH_b \approx b\log(b) + \gamma b + \frac{1}{2}$. The proof concludes with the linearity of expectation. $\qquad\square$

**Maximum Number of Samples Required with High Probability**    We now compute the number of samples $n^p_{\text{total}}$ required to recover all $b$ correct directions with high probability $1 - p$.

**Lemma B.2.** *In the same setting as Lemma 5.2, we have an upper bound $n^p_{total}$ on the number of submatrices we need to sample until we have recovered all $b$ correct direction vectors by solving the following quadratic inequality for $n^p_{total}$*

$$\frac{p}{2} \le \Phi\left(\frac{b\log(2b/p^*) - n^p_{total}q}{\sqrt{n^p_{total}q(1-q)}}\right),$$

*where $\Phi$ is the cumulative distribution function of the standard normal distribution and $p^* = p - 1 + (1 - p_{fr})^b$.*

*Proof.* At a high level, bound the number of valid directions $c^p$ we need to discover until we recover all $b$ distinct ones and then the number of submatrices $n^p_{\text{total}}$ we need to sample to obtain these $c^p$ directions, each with probability $1 - \frac{p}{2}$, before applying the union bound.

However, we first note that with probability $1 - (1 - p_{fr})^b$ we will (repeatedly) reject a correct direction due to a lack of induced sparsity and thus fail irrespective of the number of samples we draw. We thus correct our failure probability budget from $p$ to $p^* = p - 1 + (1 - p_{fr})^b$, using the union bound.

We now show how to compute the upper bound on the number of correct directions $c^p$ we need to find until we have found all $b$ distinct directions. To this end, we bound the probability of not sampling the $i^{\text{th}}$ direction $\overline{\boldsymbol{q}}_i$ after finding $c$ candidates as $p_{\neg i} = (1 - \frac{1}{b})^c \le e^{-\frac{c}{b}}$. We can then bound the probability of missing any of the $b$ directions using the union bound as $p_{\neg \text{all}} \le \sum_{i=1}^b p_{\neg i} = be^{-\frac{c^p}{b}}$. We thus obtain the minimum number $c^p$ of correct directions to find all $b$ distinct ones with probability at least $\frac{p^*}{2}$ as $c^p \ge b\log(2b/p^*)$.

We can now compute the number $n^p_{\text{total}}$ of samples required to find $c^p$ submatrices satisfying the condition of Theorem 3.3 for some $i$ with probability $1 - \frac{p}{2}$. To this end, we approximate the Binomial distribution $\mathcal{B}(n, q)$ with the normal distribution $\mathcal{N}(nq, nq(1-q))$ [35], which is generally precise if $\min(nq, n(q-1)) > 9$ [36], which holds for $b \ge 5$. We thus obtain the number of samples $n^p_{\text{total}}$

required to find $c^p$ valid directions with high probability $1 - \frac{p^*}{2}$ by solving $\frac{p^*}{2} = \Phi\left(\frac{c^p - n^p_{total}q}{\sqrt{n^p_{total}q(1-q)}}\right)$ for $n^p_{total}$ which boils down to a quadratic equation.

By the union bound, we have that the total failure probability of not finding all $b$ correct directions is at most $p$. $\qquad\square$

For a batch size of $b = 10$ and $p = 10^{-8}$, we, e.g., obtain $n \approx 4 \times 10^4$.

**Lemma 5.3.** *Under the same assumptions as in Lemma 5.1, we have an upper bound on the failure probability $p^{ub}_{fail}$ of Alg. 1 even when sampling exhaustively as:*

$$p^{ub}_{fail} \leq b\left(1 - \sum_{k=b-1}^{m} \binom{m}{k}\frac{1}{2^m}\left(1 - 0.939^{(b-1)\binom{k}{b-1}}\right)\right) + 1 - (1 - p_{fr})^b,$$

*where $p_{fr}$ is the probability of falsely rejecting a correct direction $\overline{q}'$ via our sparsity filter (Sec. 4.1).*

*Proof.* We will first compute the probability of $\frac{\partial \mathcal{L}}{\partial Z}$ not containing a submatrix $A$ satisfying the conditions of Theorem 3.3 for all $i \in \{1, \ldots, b\}$ and then the probability of us failing to discover it despite exhaustive sampling.

We observe that the number $k$ of rows in $\frac{\partial \mathcal{L}}{\partial Z}$ with a zero $i^{\text{th}}$ entry is binomially distributed with success probability $\frac{1}{2}$. For each $k \geq b - 1$, we can construct $\binom{k}{b-1}$ submatrices $A$ with an all-zero $i^{\text{th}}$ column. The probability of any such submatrix having full rank is $1 - (\frac{1}{2} - o_{b-1}(1))^{b-1} > 1 - 0.939^{b-1}$ [11, 34].

We thus have the probability of $\frac{\partial \mathcal{L}}{\partial Z}$ containing at least one submatrix $A$ with full rank and an all-zero $i^{\text{th}}$ column as $\sum_{k=b-1}^{m} \binom{m}{k}\frac{1}{2^m}\left(1 - 0.939^{(b-1)\binom{k}{b-1}}\right)$.

Using the union bound, we thus obtain an upper bound on the probability of $\frac{\partial \mathcal{L}}{\partial Z}$ not containing any submatrix $A$ with full rank and an all-zero $i^{\text{th}}$ column for all $i \in \{1, \ldots, b\}$.

To compute the probability of us failing to discover an existing submatrix despite exhaustive sampling, we first note that we have the probability $p_{fr}$ of an arbitrary column in $\frac{\partial \mathcal{L}}{\partial Z}$ being less sparse than our threshold $\tau$. Thus, with probability $1 - (1 - p_{fr})^b$ we will discard at least one correct direction due to it inducing an unusually dense column in $\frac{\partial \mathcal{L}}{\partial Z}$.

We now obtain the overall failure probability via the union bound. $\qquad\square$

## C   Deferred Algorithms

Here, we present the Algorithms COMPUTELAMBDA and GREEDYFILTER referenced in Sec. 5.1.

---
**Algorithm 2** COMPUTELAMBDA
---
1: **function** COMPUTELAMBDA($L, R, W, b, \frac{\partial \mathcal{L}}{\partial b}, \mathcal{B}$)
2: $\quad Q \leftarrow$ FIXSCALE $(\mathcal{B}, L, \frac{\partial \mathcal{L}}{\partial b})$
3: $\quad \frac{\partial \mathcal{L}}{\partial Z} \leftarrow L \cdot Q$
4: $\quad X^T \leftarrow Q^{-1} \cdot R$
5: $\quad Z = W \cdot X + (b|\ldots|b)$
6: $\quad \lambda_- \leftarrow \sum_{i,j} \mathbb{1}[Z_{i,j} \leq 0] \cdot \mathbb{1}[\frac{\partial \mathcal{L}}{\partial Z_{i,j}} = 0]$
7: $\quad \lambda_+ \leftarrow \sum_{i,j} \mathbb{1}[Z_{i,j} > 0] \cdot \mathbb{1}[\frac{\partial \mathcal{L}}{\partial Z_{i,j}} \neq 0]$
8: $\quad \lambda \leftarrow \frac{\lambda_- + \lambda_+}{m \cdot b}$
9: $\quad$**return** $\lambda$
---

**Algorithm 3** GREEDYFILT

```
 1: function GREEDYFILTER(L, R, W, b, ∂L/∂b, C)
 2:     B ← {}
 3:     while rank of B is B do
 4:         Select the sparsest vector q̄'ᵢ from C \ B
 5:         B ← B ∪ {q̄'ᵢ}
 6:         if B is not of full rank then
 7:             B ← B \ {q̄'ᵢ}
 8:         end if
 9:     end while
10:
11:     λ ← COMPUTELAMBDA (L, R, W, b, ∂L/∂b, B)
12:     while not changed do
13:         changed ← False
14:         for (q̄'ᵢ, q̄'ⱼ) in B × (C \ B) do
15:             B' ← B \ {q̄'ᵢ} ∪ {q̄'ⱼ}
16:             λ' ← COMPUTELAMBDA (L, R, W, b, ∂L/∂b, B')
17:             if λ' > λ then
18:                 B ← B'
19:                 λ ← λ'
20:                 changed ← True
21:             end if
22:         end for
23:     end while
24:     Q ← FIXSCALE (B, L, ∂L/∂b)
25:     Xᵀ ← Q⁻¹ · R
26:     return λ, X
```

Table 7: Reconstruction quality across 100 batches.

| Dataset | PSNR ↑ | LPIPS ↓ | Acc (%) ↑ | Time/Batch |
|---|---|---|---|---|
| MNIST | $99.1 \pm 13.2$ | NaN | **99** | 2.6 min |
| CIFAR-10 | $106.6 \pm 15.1$ | $1.16 \times 10^{-5} \pm 2.26 \times 10^{-4}$ | **99** | 1.7 min |
| TINYIMAGENET | $110.7 \pm 12.8$ | $1.62 \times 10^{-4} \pm 3.22 \times 10^{-3}$ | **99** | **1.4 min** |
| IMAGENET $224 \times 224$ | $125.4 \pm 11.2$ | $1.05 \times 10^{-5} \pm 9.50 \times 10^{-4}$ | **99** | 2.1 min |
| IMAGENET $720 \times 720$ | $\mathbf{125.6 \pm 8.1}$ | $\mathbf{8.08 \times 10^{-11} \pm 3.05 \times 10^{-3}}$ | **99** | 2.6 min |

# D  Dataset Licenses

In this work, we use the commonly used MNIST [13], CIFAR-10 [14], TINYIMAGENET [15] and IMAGENET [16] image datasets. No information regarding licensing has been provided on their respective websites. Further, we use Adult tabular dataset under the Creative Commons Attribution 4.0 International (CC BY 4.0) license.

# E  Deferred Experiments

## E.1  Main Results with Error Bars

In this section, we provide the results from our main experiment in Table 3, alongside 95% confidence intervals.

## E.2  Experiments on Label-Heterogeneous Data

In this section, we provide experiments on heterogeneous client data. In particular, we look at the extreme case where each client has data only from a single class. As label repetition makes optimization-based attacks harder [1, 23, 7], the results presented in Table 8 for the TinyImageNet

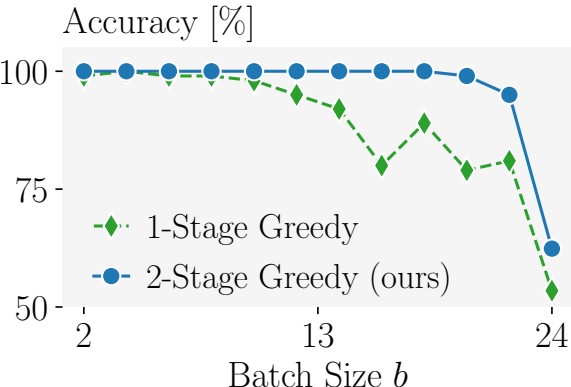

Figure 6: Effect of the second stage of our reconstruction algorithm discussed in Sec. 4.2, depending on the batch size $b$.

dataset show another advantage of our algorithm, namely, SPEAR works regardless of the label distribution, providing even better reconstruction results compared to Table 3 for single-label batches.

Table 8: Mean reconstruction quality metrics across 100 batches for batches only containing samples from only one class in the same setting as Table 3.

| Dataset | PSNR ↑ | SSIM ↑ | MSE ↓ | LPIPS ↓ | Acc (%) ↑ |
|---|---|---|---|---|---|
| TINYIMGNET | 127.7 | 0.999717 | $4.80 \times 10^{-6}$ | $10.36 \times 10^{-5}$ | 98 |

### E.3 Effectivness of our 2-Stage Greedy Algorithm

In this section, we compare reconstruction success rate (accuracy) with and without the second stage of our greedy algorithm discussed in Sec. 4.2 in Fig. 6. We observe that the second stage filtering becomes increasingly important for larger batch size $b$.

### E.4 Effect of Training on SPEAR

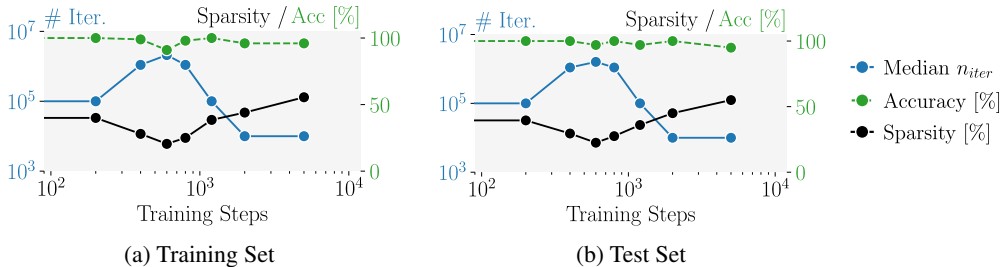

(a) Training Set        (b) Test Set

Figure 7: Effect of training (on MNIST) on the effectiveness of SPEAR at a batch size of $b = 10$ evaluated on the MNIST training (a) and test (b) sets.

In this section, we demonstrate how training effects SPEAR's performance. To this end, we train a network on MNIST and evaluate SPEAR periodically during training both on the train and test datasets, visualizing results in Fig. 7. We observe that SPEAR performance is very similar between the two datasets we evaluate on. Further, we see that SPEAR performs very well on trained networks, with the number of required steps by the algorithm being even lower those those on untrained networks. However, if the minimum column sparsity of $\frac{\partial \mathcal{L}}{\partial \mathbf{Z}}$ drops significantly, as is the case for the checkpoints around 1000 training steps in the illustrated run. SPEAR's performance drops slightly.

### E.5 Failure Probabilities

In this section, we validate experimentally our theoretical results on SPEAR's failure rate for several batch sizes $b$ (Lemma 5.3). As this requires exhaustive sampling of all $\binom{m}{b-1}$ submatrices of $L$ we only consider small batch sizes $b \leq 10$ and networks $m \leq 40$. We show the results in Fig. 8 where we observe that the empirical failure probability (blue) with $95\%$ Clopper-Pearson confidence bounds generally agrees with the analytical approximation (solid line) and always lies below the analytical upper bound (dashed line). We conclude that in most settings, the number of required samples rather than complete failure is the limiting factor for SPEAR's performance.

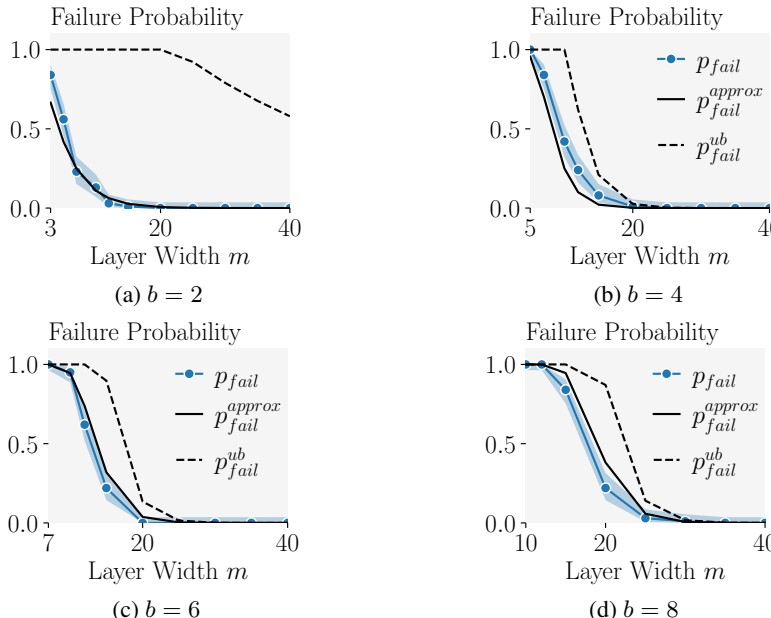

Figure 8: Empirical failure probability (blue) with $95\%$ Clopper-Pearson confidence bounds (shaded blue) compared to the analytical upper bound (dashed line) and approximation (solid line) of the failure probability for different batch sizes $b$.

### E.6 Results under DPSGD

In this section, we show experimental results on reconstructing images from gradients defended using DP-SGD [28]. In Table 9, we report results on the TINYIMAGENET dataset, $b = 20$, with noise levels $\sigma \leq 1.0 \times 10^{-4}$ and gradient clipping that constrains the $\ell_2$ norm of the composite gradient vector, combining the gradients of all layers, to a maximum value of $C \in [1, 2]$. We chose the maximum value $\sigma$ to be close to median gradient magnitude of the first linear layer which in our experiments was also $\approx 1.0 \times 10^{-4}$. We chose the range for $C$ such that for the upper bound 2, most individual input gradients are not clipped, while for the lower bound 1 almost all are.

**Adapting SPEAR to Noisy Gradients** In the experiments presented in Table 9, we make several adjustments to SPEAR to better handle the noise added by DPSGD. First, we apply looser thresholds in our sparsity filtering at Line 6 in Alg. 1 to account for the noise added to the sparse entries of $\frac{\partial \mathcal{L}}{\partial Z}$. To account for the imperfect reconstructions in this setting, we also perform our early stopping (Line 9 in Alg. 1) when the sparsity matching coefficient $\gamma$ reaches a lower value than 1. Further, we sample matrices $L_A$ of larger size $(b + 1 \times b)$ to increase the numerical stability of our solutions under noise. While sampling larger $L_A$ is more computationally expensive, as $b + 1$ instead of $b - 1$ entries in $A$ are required to be correctly sampled as 0, the resulting directions $q_i$ are more numerically stable as they are obtained as a solution of an overdetermined system of linear equations. Note that if $A$ is assumed to be of rank $b - 1$, Theorem 3.3 remains valid for these larger matrices $L_A$. Finally, due to our looser sparsity filtering described above we encounter more incorrect directions $\overline{q}_i$. We tackle this issue by only keeping $\overline{q}_i$ that correspond to matrices $L_A$ of rank exactly $b - 1$. Under our

assumption in Theorem 3.3, those are exactly the vectors $\overline{q}_i$ that correspond to $A$ of the correct rank $b - 1$. Note that we apply these changes only for $\sigma > 0$.

**Invariance to Gradient Clipping**  In Table 9, we observe that the quality of our reconstructions is not affected by the clipping constant $C$. This is not a coincidence, but rather a mathematical fact. To see this, note that the observed gradients w.r.t. $W$ under clipping are given by:

$$\frac{\dot{\partial}\mathcal{L}}{\partial W} = \sum_{i=1}^{b} c_i \frac{\partial\mathcal{L}}{\partial W_i} = \sum_{i=1}^{b} c_i \frac{\partial\mathcal{L}}{\partial Z_i} X_i,$$

where $c_i \in \mathbb{R}$ are the unknown to the attacker factors applied by the clipping procedure to each individual input gradient $\frac{\partial\mathcal{L}}{\partial W_i}$. One can adapt the proof to Theorem 3.1, to show that $\frac{\dot{\partial}\mathcal{L}}{\partial W} = \frac{\dot{\partial}\mathcal{L}}{\partial Z} X^T$, where we define $\frac{\dot{\partial}\mathcal{L}}{\partial Z_i}$ to be the clipped gradient w.r.t $Z$, consisting of the columns $\frac{\dot{\partial}\mathcal{L}}{\partial Z_i} = c_i \frac{\partial\mathcal{L}}{\partial Z_i}$. We also observe that one can adapt Theorem 3.4 to work directly on the clipped gradients as well, resulting in the formula $\frac{\dot{\partial}\mathcal{L}}{\partial b} = \frac{\dot{\partial}\mathcal{L}}{\partial Z}\begin{bmatrix}1\\ \vdots\\ 1\end{bmatrix}$ for the clipped gradient w.r.t. $b$. The formula follows from the observation that in our setting the same clipping factor $c_i$ is applied to the gradients of each layer, including $\frac{\partial\mathcal{L}}{\partial b_i}$ and $\frac{\partial\mathcal{L}}{\partial W_i}$. By applying the rest of the theoretical results of the paper without change but on clipped gradients $\frac{\dot{\partial}\mathcal{L}}{\partial Z}$, instead of the original unclipped gradients $\frac{\partial\mathcal{L}}{\partial Z}$, we conclude that SPEAR is directly applicable on the clipped client gradient and that applying it on those still recover the true input matrix $X$ without the need of knowing the clipping constants $c_i$.

**Robustness to Noise**  From Table 9, we observe that SPEAR is very robust to noise. We emphasize in particular that even when noise of similar size to the size of the gradients in expectation is applied, we still obtain a reconstruction with PSNR $> 28$. This is similar to the PSNR of 29.3 that Geiping et al. [1] achieves *without any noise* which is commonly considered unacceptable information leakage. These experiments suggest that to efficiently defend against SPEAR using noise, one needs to apply such high magnitudes that training will likely be significantly impeded.

# F   SPEAR under FedAvg Updates

In this section, we first demonstrate theoretically that SPEAR can be generalized to attack FedAvg [4] client updates, and then present empirical results confirming that SPEAR is indeed very effective under FedAvg protocols with different number of epochs $\mathcal{E}$, local client learning rates $\eta$, and, even works, when mini-batches of size $b_{\text{mini}}$ are used.

**Generalizing SPEAR to FedAvg Updates**  Assuming that a client uses all of its data points, $X$, in each local gradient step of the FedAvg protocol, i.e. $b_{\text{mini}} = b$, the client computes and subsequently shares with the server the following updated linear layer weights:

$$W^{\mathcal{E}} = W^0 - \eta \sum_{e=1}^{\mathcal{E}} \frac{\partial\mathcal{L}}{\partial W^e} = W^0 - \eta \sum_{e=1}^{\mathcal{E}} \frac{\partial\mathcal{L}}{\partial Z^e} \cdot X^T = W^0 - \eta \left( \sum_{e=1}^{\mathcal{E}} \frac{\partial\mathcal{L}}{\partial Z^e} \right) \cdot X^T,$$

where $W^0$ is the global model sent by the server, $W^e$ represent the local client weights after $e$ client epochs, and $\frac{\partial\mathcal{L}}{\partial W^e}$ and $\frac{\partial\mathcal{L}}{\partial Z^e}$ are the weight and output gradients at epoch $e$.

We empirically observe that sparsity patterns of the different local gradients $\frac{\partial\mathcal{L}}{\partial Z^e}$ are usually similar. This is expected as these patterns correspond to the ReLU activation patterns for the layer outputs $Z^e$ (see Sec. 3.2) at different local steps which are computed on the same data $X$ and with similar weights $W^e$. As the sparsity patterns for the individual gradients are similar, their sum $\sum_{e=1}^{\mathcal{E}} \frac{\partial\mathcal{L}}{\partial Z^e}$ also shares this sparsity pattern and is, thus, also sparse. As the server knows $W^0$ and it can subtract it from the client's shared weights $W^{\mathcal{E}}$ and apply Theorem 3.3, as before, on the sparse matrix $\sum_{e=1}^{\mathcal{E}} \frac{\partial\mathcal{L}}{\partial Z^e}$ to obtain the corresponding matrix $Q$ and client data $X$. We note that while our sparsity matching coefficient $\sigma$ will typically not reach 1 for the final reconstruction in this setting, as there is some mismatch between the sparsity patterns of the different output gradients $\frac{\partial\mathcal{L}}{\partial Z^e}$, we have found that SPEAR remains practically effective regardless.

Table 9: Reconstruction quality across 100 batches of size $b = 20$ computed on TINYIMAGENET for gradients computed with DPSGD [28] with different noise levels $\sigma$ and gradient clipping levels $C$.

| Method | $C$ | $\sigma$ | PSNR $\uparrow$ | Acc (%) $\uparrow$ |
|---|---|---|---|---|
| Geiping et. al [1] | 0.00 | 0 | 29.3 | 100 |
| SPEAR (Ours) | 1.00 | 0 | 118.2 | 100 |
| SPEAR (Ours) | 1.25 | 0 | 118.1 | 100 |
| SPEAR (Ours) | 1.50 | 0 | 118.5 | 100 |
| SPEAR (Ours) | 1.75 | 0 | 118.7 | 100 |
| SPEAR (Ours) | 2.00 | 0 | 118.0 | 100 |
| SPEAR (Ours) | 1.00 | $5.0 \times 10^{-6}$ | 38.6 | 99 |
| SPEAR (Ours) | 1.25 | $5.0 \times 10^{-6}$ | 40.4 | 98 |
| SPEAR (Ours) | 1.50 | $5.0 \times 10^{-6}$ | 41.9 | 98 |
| SPEAR (Ours) | 1.75 | $5.0 \times 10^{-6}$ | 42.2 | 97 |
| SPEAR (Ours) | 2.00 | $5.0 \times 10^{-6}$ | 42.0 | 96 |
| SPEAR (Ours) | 1.00 | $1.0 \times 10^{-5}$ | 38.2 | 99 |
| SPEAR (Ours) | 1.25 | $1.0 \times 10^{-5}$ | 40.0 | 98 |
| SPEAR (Ours) | 1.50 | $1.0 \times 10^{-5}$ | 38.5 | 99 |
| SPEAR (Ours) | 1.75 | $1.0 \times 10^{-5}$ | 39.2 | 99 |
| SPEAR (Ours) | 2.00 | $1.0 \times 10^{-5}$ | 39.6 | 99 |
| SPEAR (Ours) | 1.00 | $5.0 \times 10^{-5}$ | 32.3 | 97 |
| SPEAR (Ours) | 1.25 | $5.0 \times 10^{-5}$ | 33.5 | 98 |
| SPEAR (Ours) | 1.50 | $5.0 \times 10^{-5}$ | 34.4 | 99 |
| SPEAR (Ours) | 1.75 | $5.0 \times 10^{-5}$ | 34.6 | 100 |
| SPEAR (Ours) | 2.00 | $5.0 \times 10^{-5}$ | 34.1 | 100 |
| SPEAR (Ours) | 1.00 | $1.0 \times 10^{-4}$ | 29.7 | 98 |
| SPEAR (Ours) | 1.25 | $1.0 \times 10^{-4}$ | 29.3 | 97 |
| SPEAR (Ours) | 1.50 | $1.0 \times 10^{-4}$ | 29.9 | 99 |
| SPEAR (Ours) | 1.75 | $1.0 \times 10^{-4}$ | 29.4 | 98 |
| SPEAR (Ours) | 2.00 | $1.0 \times 10^{-4}$ | 28.7 | 95 |

We note that SPEAR can be even be generalized to FedAvg protocols that use random mini-batches $\boldsymbol{X}^e$ of size $b_{\text{mini}} < b$ sampled from $\boldsymbol{X}$ at each local step. This is the case, as each local client gradient $\frac{\partial \mathcal{L}}{\partial \boldsymbol{W}^e} = \frac{\partial \mathcal{L}}{\partial \boldsymbol{Z}^e}(\boldsymbol{X}^e)^T$, can be represented as $\overline{\frac{\partial \mathcal{L}}{\partial \boldsymbol{Z}^e}}\boldsymbol{X}^T$, where $\overline{\frac{\partial \mathcal{L}}{\partial \boldsymbol{Z}^e}}$ is derived from $\frac{\partial \mathcal{L}}{\partial \boldsymbol{Z}^e}$ by adding 0 columns at batch positions corresponding to batch elements not in $\boldsymbol{X}^e$. Importantly, as $\overline{\frac{\partial \mathcal{L}}{\partial \boldsymbol{Z}^e}}$ only adds 0 columns to $\frac{\partial \mathcal{L}}{\partial \boldsymbol{Z}^e}$, the sparsity of $\overline{\frac{\partial \mathcal{L}}{\partial \boldsymbol{Z}^e}}$ can only increase, allowing to conclude that $\sum_{e=1}^{\mathcal{E}} \overline{\frac{\partial \mathcal{L}}{\partial \boldsymbol{Z}^e}}$ remains sparse, and, thus, Theorem 3.3 can still be applied to it.

**Experiments with FedAvg Updates**   Next, we show empirically the effectiveness of SPEAR for FedAvg updates. In Table 10, we show the results of attacking clients with $b = 20$ datapoints from the TINYIMAGENET dataset for different number of local client epochs $\mathcal{E}$. We observe that even for $\mathcal{E} = 50$ gradient steps we recover data from most batches, with quality similar to the quality achieved when attacking individual gradients. This is expected as Theorem 3.3 still holds, as described in the previous paragraph. The slight dip in the fraction of reconstructed batches for larger number of steps $\mathcal{E}$ can be attributed to some client batches inducing larger discrepancy between the sparsity patterns of $\frac{\partial \mathcal{L}}{\partial \boldsymbol{Z}^e}$ compared to others, resulting in their sum being much less sparse. Further, Table 10 also shows that SPEAR can attack client updates that take $b/b_{\text{mini}} = 4$ local steps per epoch for $\mathcal{E} = 20$ epochs. Interestingly, while a total of 80 gradient steps are taken in this scenario the results are closer to the $b_{\text{mini}} = 20, \mathcal{E} = 20$ setting, instead of the $b_{\text{mini}} = 20, \mathcal{E} = 50$ setting. This can be explained by the increased sparsity of the individual expanded gradients $\overline{\frac{\partial \mathcal{L}}{\partial \boldsymbol{Z}^e}}$.

Finally, we experiment with different local client learning rates $\eta$ and show the results in Table 11. We observe that even for large learning rates SPEAR still recovers its inputs well, showing that while the individual weights $\boldsymbol{W}^e$ can change a lot, their induced sparsity on $\frac{\partial \mathcal{L}}{\partial \boldsymbol{Z}^e}$ remains consistent.

# G   Additional Visualisations

In this section we present additional visualisations of the reconstructions obtained by SPEAR. First, in Fig. 9 we show an extended comparison between the images recovered by our method and Geiping

Table 10: Reconstruction quality across 100 FedAvg client updates computed on TINYIMAGENET batches of size $b = 20$ for different number of epochs $\mathcal{E}$ and different mini batch sizes $b_{\text{mini}}$.

| $\eta$ | $\mathcal{E}$ | $b_{\text{mini}}$ | PSNR ↑ | Acc (%) ↑ |
|---|---|---|---|---|
| 0.01 | 1 | 20 | 97.8 | 97 |
| 0.01 | 5 | 20 | 103.9 | **100** |
| 0.01 | 10 | 20 | 106.7 | 99 |
| 0.01 | 20 | 20 | **108.90** | 98 |
| 0.01 | 50 | 20 | 104.9 | 90 |
| 0.01 | 20 | 5 | 106.7 | 97 |

Table 11: Reconstruction quality across 100 FedAvg client updates computed on TINYIMAGENET batches of size $b = 20$ for different local client learning rates $\eta$.

| $\eta$ | $\mathcal{E}$ | $b_{\text{mini}}$ | PSNR ↑ | Acc (%) ↑ |
|---|---|---|---|---|
| 0.1 | 5 | 20 | **119.3** | 95 |
| 0.01 | 5 | 20 | 103.9 | **100** |
| 0.001 | 5 | 20 | 85.5 | **100** |

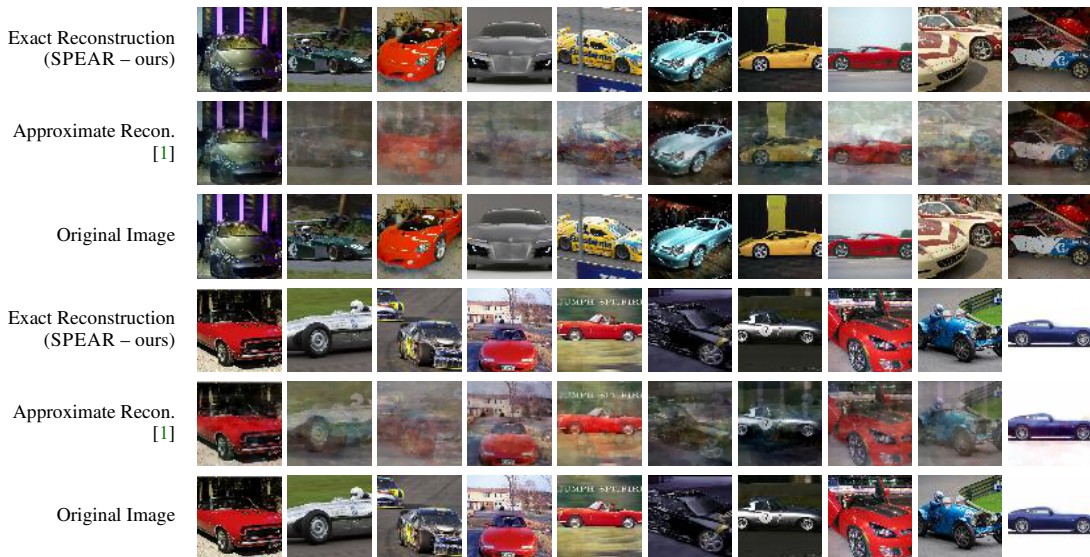

Exact Reconstruction (SPEAR – ours)

Approximate Recon. [1]

Original Image

Exact Reconstruction (SPEAR – ours)

Approximate Recon. [1]

Original Image

Figure 9: The reconstructions of all images from Fig. 1, reconstructed using our SPEAR (top) or the prior state-of-the-art Geiping et al. [1] (mid), compared to the ground truth (bottom).

et al. [1] on the TINYIMAGENET batch first shown in Fig. 1. In Fig. 9 we operate in the same setting as Table 8, namely batches of only a single class. We observe that while some images are reconstructed well by Geiping et al. [1], most of the images are of poor visual quality, with some even being hard to recognize. In contrast, all of our reconstructions are pixel perfect. This in particular also means, that SPEAR's reconstructions improve in fine-detail recovery even upon the well recovered images of Geiping et al. [1]. This is expected as our attack is exact (up numerical errors).

Further, to show the results in Fig. 9 are representative, in Fig. 10–12 we provide additional visualizations of the reconstructions obtained by SPEAR corresponding to the $10^{\text{th}}$, $50^{\text{th}}$, and $90^{\text{th}}$ percentiles of the PSNRs obtained in the TINYIMAGENET experiment reported in Table 3. We observe that only 1 sample has visual artefacts for the $10^{\text{th}}$ percentile batch (top left image in Fig. 10) and that the $50^{\text{th}}$ and $90^{\text{th}}$ percentile batches contain only perfect reconstructions. We theoreticize that the visual artefact in Fig. 10 is a result of a numerical instability issue and that using $\boldsymbol{L}_A$ of bigger size as described in App. E.6 one could further alleviate it in exchange of additional computation.

Finally, we demonstrate what happens to SPEAR reconstructions in the rare case when the algorithm fails to recover all correct directions $\bar{q}_i$ from the batch gradient. In Fig. 13, we show the only such batch for the TINYIMAGENET experiment reported in Table 3. The batch has 2 wrong directions and

still achieves an average PSNR of 91.2 (the worst PSNR obtained in this experiment), which is still much higher compared to prior work. Further, all but 2 images are affected by the failure.

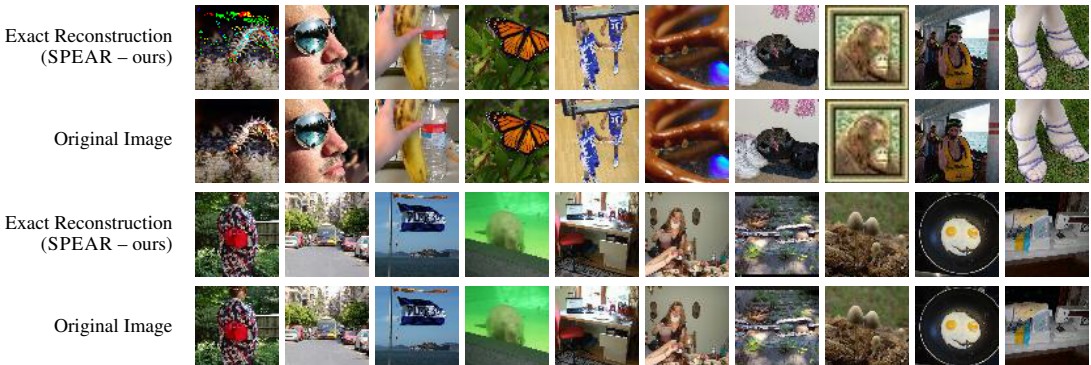

Figure 10: Visualisation of the images reconstructed by SPEAR from the batch whose PSNR is at the 10th percentile based on the set of 100 TINYIMAGENET reconstructions reported in Table 3.

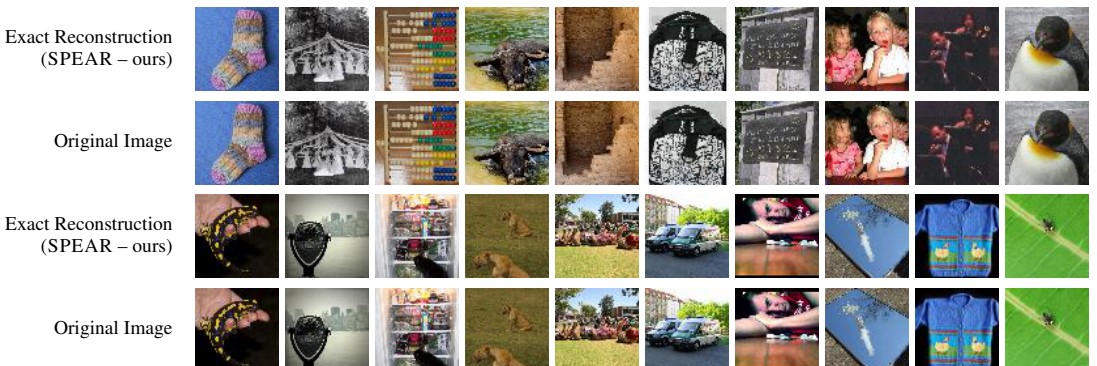

Figure 11: Visualisation of the images reconstructed by SPEAR from the batch whose PSNR is at the 50th percentile based on the set of 100 TINYIMAGENET reconstructions reported in Table 3.

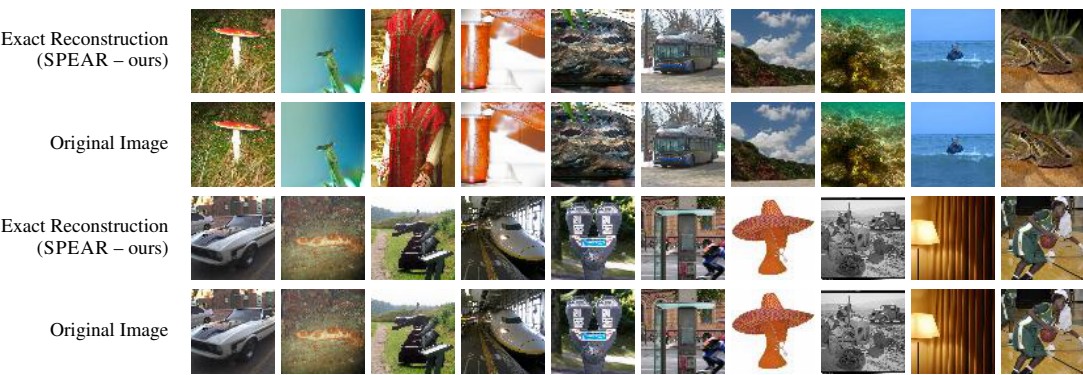

Figure 12: Visualisation of the images reconstructed by SPEAR from the batch whose PSNR is at the 90th percentile based on the set of 100 TINYIMAGENET reconstructions reported in Table 3.

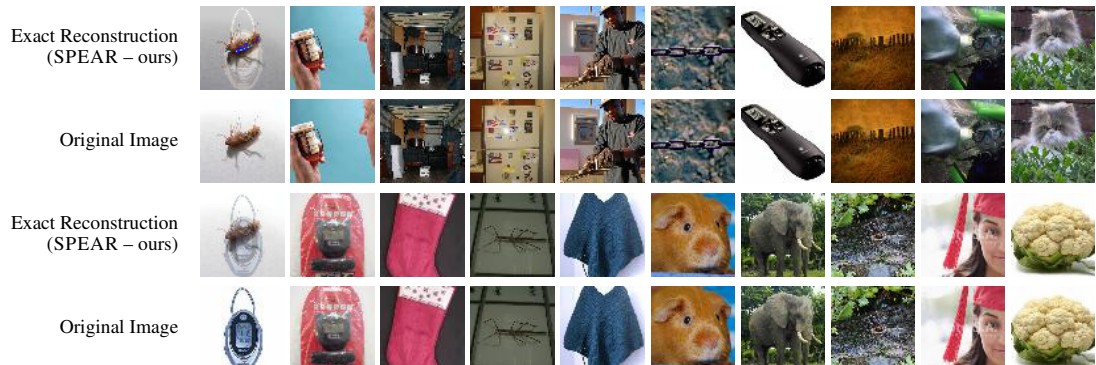

Figure 13: Visualisation of the images reconstructed by SPEAR from the only batch from the 100 TINYIMAGENET reconstructions reported in Table 3, where not all recovered directions $\overline{q}'_i$ are correct. SPEAR recovered 2/20 wrong directions, resulting in the left most images being wrongly recovered.

