# OpenReview forum: "SPEAR: Exact Gradient Inversion of Batches in Federated Learning"
_NeurIPS.cc/2024/Conference — NeurIPS 2024 poster_

### Official Review · Reviewer_Xnvh · 2024-07-10

**Soundness:** 4
**Presentation:** 3
**Contribution:** 3
**Rating:** 6
**Confidence:** 3

**Summary:**

This paper presents a novel method to recover batched input data from gradients in fully-connected networks. Based on a detailed analysis of the rank of gradients, it utilizes matrix decomposition to first figure out the batch size and then recover the input tensor through sparsity. Good comparisons with previous methods have demonstrated the performance of such a method.

**Strengths:**

- Good writing. The structure is well-organized.
- It is of great value to discover the mathematical relationships within the gradient matrixes, and then use matrix decomposition to solve this problem.
- Comparisons with previous methods, both in the image and tabular data domains, make this work convincing.

**Weaknesses:**

Please refer to the Questions part.
Minor problem: It is recommended that the work should be double-checked in case of typos or compiling errors.
- Line 79 typo
- Line 125 typo
- Line 149 abnormal block

**Questions:**

- It seems this work relies highly on the sparsity. What if the activation function is sigmoid, leakyReLU, tanh, etc? These activations do not directly filter out negative inputs and return 0s, which may affect the sparsity property.
- For experiments in 6.2, are the image labels known for the reconstruction ahead of time?
- Why is it that this method may fail when batchsize is larger than 25? In my personal view, more insights could be covered about this direction in the failure probability part.

**Limitations:**

The proposed method only works for FCNs.

---

> ### Author Rebuttal · Authors · 2024-08-07
>
> We thank the reviewer for their very positive review. We are particularly happy to read that the reviewer assesses our contribution to be of great value and finds the comparison to related work convincing. We will include all proposed writing suggestions. We now address all questions of the reviewer.
>
> **Q1: Can SPEAR handle activation functions different from ReLU?**
>
> We direct the reviewer to our previous answer to question **Q2** in the general response. In this work, we focus on the most common ReLU activation as it is the most relevant one in practice. Further, we believe that transferring our insights to other activations is a valuable item for future work.
>
> **Q2: For experiments in 6.2, are the image labels known for the reconstruction ahead of time?**
>
> No, we do not require the knowledge of labels at all. We do want to emphasize that the only prior we utilize is the ReLU-induced sparsity.
>
> **Q3: Why can SPEAR fail when the batch size is larger than 25? Is this handled by the probability of failure analysis?**
>
> We refer the review to our answer to Q1 in the main response. As outlined there, the failure probability for reasonable network widths is not a significant bottleneck for the batch sizes we experimented with. Instead, the bottleneck is the number of submatrices $L_A$ one needs to sample to recover all the correct directions $\overline{q}$, which is analyzed in Lemma 5.2. In practice, we limit this number to fit in a reasonable time limit and we report a failure otherwise. We emphasize that SPEAR, in theory, is capable of recovering inputs for batch sizes $> 25$ provided enough time. As outlined in the general response to Q1, one can use approximate reconstruction methods to guide the search for those submatrices, lifting the restriction, and allowing exact reconstructions for batch sizes as large as 100. Finally, we provide experiments in Appendix E5, showing that the predicted failure probabilities agree well with empirical results for small to moderate layer widths.
>
> We are happy to discuss any remaining or follow-up questions the reviewer might have.

---

### Official Review · Reviewer_LPF2 · 2024-07-12

**Soundness:** 3
**Presentation:** 3
**Contribution:** 2
**Rating:** 5
**Confidence:** 4

**Summary:**

This paper studied the gradient inversion problem in federated learning. In particular, an honest-but-curious server follows the federated training protocol but aims to infer private information from gradients collected from clients. The paper proposed a novel approach, SPEAR, that exploits the low-rank and sparsity properties of gradients corresponding to linear layers with ReLU activations, and recovers input data of batch size greater than 1 almost perfectly with high probability. The algorithm is a combination of matrix decomposition and a optimisation-based filtering procedure. The performance of the algorithm is explained by theoretical analysis and empirical evaluations, which is significantly better than previous methods that struggle to deal with large batch sizes.

**Strengths:**

- Recovering batched images from gradients has been a challenging task. The algorithm introduced in this paper is built on insightful observations on the gradients and solid theoretical justifications, which can inspire further work in this area.

- The proposed algorithm works well empirically, under necessary requirements such as fully connected networks and moderate batch size. Sufficient ablations are provided to evaluate different aspects of the algorithm.

- The exposition of the paper is nicely structured.

**Weaknesses:**

- The effectiveness of the algorithm relies on the assumption that the first layer of the network is a linear layer (while for linear layers in the middle or at the end of the network the algorithm can only recover the intermediate input to those layers, not the original images). However, this is not the case for many popular convolutional networks for image tasks. The application of the algorithm is restricted and it can be easily defended in practice.

- The running time of the algorithm is exponential in the batch size, and the success probability may drop significantly for large batch size. Moreover, it is not clear if the algorithm is effective for high-resolution datasets like ImageNet.

**Questions:**

- When the batch size increases, is there a trade-off scheme that sacrifices the fidelity of recovered images but reduces the running time?

- Can the algorithm deal with normalization layers?

- Can the algorithm be adapted to convolutional networks and (vision) transformers?

**Limitations:**

The paper discussed limitations.

---

> ### Author Rebuttal · Authors · 2024-08-07
>
> We thank the reviewer for their positive review. We are happy to read that they find our contribution insightful and inspiring future work and credit our significantly better performance and theoretical analysis. Further, we are glad the reviewer deems the paper nicely structured.
>
> **Q1: Can the exponential sample complexity of SPEAR be improved? Is the probability of failure a bottleneck in practice?**
>
> Yes, the exponential sample complexity can be addressed, as we discuss in the general response (Q1). Specifically, there we show that SPEAR can leverage optimization-based methods to circumvent the exponential sampling-based information retrieval to scale to much larger batch sizes while still providing (almost) perfect reconstruction, and we clarify that the failure probability is not currently a bottleneck (see also Figure 3 in the main paper).
>
> **Q2: Is the algorithm effective for high-resolution datasets like ImageNet?**
>
> As highlighted in the abstract, SPEAR reconstructs batches of sizes up to 25 on ImageNet exactly. We provide more corresponding results in Section 6.2 (Main Results). Specifically, we run experiments on the commonly used resolution of 224x224 for ImageNet and also on a 720x720 high-resolution setting to showcase our scalability even further. We note that this is, as shown in Table 3, at the very moderate cost of a runtime increase from 2.1 minutes to 2.6 minutes per batch. This is significantly faster than prior work as shown in Table 1, where we compare various works on ImageNet using a resolution of 256x256. Could the reviewer clarify the question in case we misunderstood it?
>
> **Q3: Does the effectiveness of the algorithm rely on the assumption that the first layer of the network is a linear layer?**
>
> No. As discussed in Q3 of the main response, SPEAR can be efficiently applied to any linear layer that precedes or follows ReLU activations to obtain that layer’s inputs. Even when the attacked layer is in the middle of the network the intermediate inputs recovered via SPEAR pose an increased risk to the privacy of the client data. We demonstrate the practicality of using SPEAR on linear layers in the middle of the network by attacking VGG16, as shown in Table 1 in the rebuttal PDF. See Q3 of the main response for further details.
> Further, we emphasize that our method is a significant theoretical advancement over prior work, providing novel insights into the gradient leakage problem as acknowledged by the reviewers. We focus on ReLU with linear layers to open the field for further research. There is already very promising future research building on the insights gained in this work. See our answer to Q6.
> Finally, for tabular datasets, networks starting with a linear layer are common. In this setting SPEAR outperforms all prior work by a wide margin in terms of input reconstruction quality and speed, particularly when no further priors are given.
>
> **Q4: When the batch size increases, is there a trade-off scheme that sacrifices the fidelity of recovered images but reduces the running time?**
>
> Yes, the threshold we use to test if an entry in $L \overline q_i$ is 0 can be tuned. As we use this test to filter out wrong proposal directions (see Section 4.1), high thresholds can permit directions computed in a numerically unstable way (e.g. due to applying Theorem 3.4 on matrices $L_A$ that are not so well-conditioned), leading to lower reconstruction quality but faster runtime. While tuning this threshold can yield a noticeable speedup, it does not address the exponential nature of our algorithm
> However, if one would like to resolve the exponential runtime of SPEAR, we propose a combination with an optimization-based method to get priors on the signs of $Z$. Please see our reply to question Q1 in the main response, where we discuss in detail how this allows SPEAR to scale to significantly larger batch sizes while retaining exact reconstruction.
>
> **Q5: Can the algorithm deal with normalization layers?**
>
> As Batch-Norm layers compute batch statistics at training time and then use these statistics to normalize the activations, they entangle the gradients of all batch elements. This makes the analysis of the gradient structure significantly more complex and prevents us from directly applying SPEAR as now the output of the BN rather than the linear layer has sparse gradients. On the other hand, this normalization and rescaling allows us to compute a very precise estimate of the expected sparsity for the ReLU inputs $Z$ and thus improve our filtering described in Section 4. While we have promising initial results addressing this issue, SPEAR requires significant non-trivial adaptations for BN-layers, which we believe is an interesting item for future work.
>
> **Q6: Can the algorithm be adapted to convolutional networks and (vision) transformers?**
>
> Yes, there is recently published work on Transformers relying on insights from this work, that also significantly outperforms all prior work. We contacted the chair to share this work with you. For applications to convolutional networks, please see our reply to Q3 in the main response.

---

### Official Review · Reviewer_38cL · 2024-07-13

**Soundness:** 3
**Presentation:** 3
**Contribution:** 3
**Rating:** 7
**Confidence:** 3

**Summary:**

This paper introduces a novel approach to input reconstruction for neural networks, focusing specifically on linear layers followed by ReLU activations.
The key aspects of this method are:

Low-rank decomposition: The authors leverage low-rank matrix decomposition techniques to simplify the representation of the linear layer's weight matrix.

ReLU sparsity exploitation: The method takes advantage of the sparsity induced by the ReLU activation function following the linear layer. This sparsity provides additional constraints that aid in the reconstruction process.

Efficient disaggregation algorithm: A core contribution of the paper is an efficient algorithm for searching the columns of the disaggregation matrix Q. This matrix plays a crucial role in the reconstruction process.

Input reconstruction: By combining these elements, the authors demonstrate a method to reconstruct input examples from the output of a linear layer followed by a ReLU activation.

**Strengths:**

Its a really interesting idea to reconstruct from gradients and provide a mathematical explanation of previous work [1] (from the paper’s citation) and this work.

Proof is clean and precise. Also discuss many settings, like layer width and batch size. It also provides a theory bound for failure probability in reconstruction attacks.

The reconstruction result and reuse quality for reconstruction images are really good.

[1] Kariyappa, Sanjay, et al. "Cocktail party attack: Breaking aggregation-based privacy in federated learning using independent component analysis." International Conference on Machine Learning. PMLR, 2023.

**Weaknesses:**

It performs better than [1] in the reconstruction result, but the batch size would be a bottleneck since the method is bound by batch size to resolve, and [1] can reach out to batch size > 64 easily.

Interesting idea but miss some experiments and discussion:

1. It only provides experiments run on 1st linear layer setting. However, in many NN designs,  linear layers are usually after the conv layer or before softmax. Could you provide a more detailed discussion about using your method for linear layer after conv layers in more practical setting? like [9]’s section 4

2. In the supplement code section, it only provides experiments based on relu1 and fc1, Could you run quick experiments and report performance on different/later fc layers?

If the result is limited to the first fc layer, then it has some interesting innovation in method design but is not really practical.

Also, if authors can provide more no cherry-picking visualization results on different fc layers or fc after conv for different datasets in the appendix,?

For E6 DPSGD part, usually sigma = 1*10-4 is considered to be really large eps and too little noise, which is not considered practical privacy protection. For DPSGD on  linear layer NN, sigma = 0.01 with some reasonable setting can be considered as eps = 80–100 ( I might calculate wrong but its around that number), which is still not useful for DPSGD.

[1] Kariyappa, Sanjay, et al. "Cocktail party attack: Breaking aggregation-based privacy in federated learning using independent component analysis." International Conference on Machine Learning. PMLR, 2023.

**Questions:**

See above.

**Limitations:**

See above.

---

> ### Author Rebuttal · Authors · 2024-08-07
>
> We thank the reviewer for their positive review and are glad to read they found our contribution really interesting, the proofs clean and precise and the reconstruction quality really good.
>
> **Q1: Can SPEAR scale to batch sizes greater than 64?**
> Yes, we address this in the general response (Q1). Specifically, we show that SPEAR can leverage optimization-based methods to scale to significantly larger batch sizes while still providing (almost) perfect reconstruction.
>
> **Q2: Can SPEAR handle a linear layer that is preceded by a conv layer?**
> Yes. We added an experiment showing perfect reconstruction results for VGG16 and $b=20$ in Table 1 in the rebuttal PDF. Please refer to Q3 in the general response for more information about this experiment.
>
> **Q3: Can you provide experiments, where SPEAR reconstructs inputs to linear layers that are not the first one?**
> Yes. We consider a 6-layer network with 400 neurons in each layer and a batch size of 20 and report the mean absolute error (MAE) to the original activations in Table 2 in the rebuttal PDF. Our experiments indicate that we can recover almost all inputs to all layers (almost) perfectly. Please refer to Q3 in the main rebuttal for more details about this experiment.
>
> **Q4: Are the provided visualizations representative? Can the authors provide more visualizations of the SPEAR’s results?**
>
> We want to clarify that the visualizations provided in Figure 1 of the main paper are a subsample of the visualizations provided in Figure 9 in Appendix G, where we visualize the corresponding full batch. The batch in Figure 9 is representative as it was randomly chosen and has a PSNR similar to the overall PSNR mean. To reaffirm this point, we provide further batch visualizations in Figures 1-3 in the rebuttal PDF. These visualizations are in the same setting as Figure 1 of the main paper, but their corresponding batches are chosen to represent the $10^{\text{th}}$,$50^{\text{th}}$, and $90^{\text{th}}$ percentiles of the obtained PSNRs in this experiment. We observe that only 1 sample is not perfectly reconstructed for the $10^{\text{th}}$ percentile batch (which we show in its corresponding figure) and that the $50^{\text{th}}$ and $90^{\text{th}}$ percentile batches contain only perfect reconstructions.
>
> Further, we note that Figure 10 in Appendix G visualizes the batch on which SPEAR performed the worst on TinyImageNet. Manual inspection showed that its poor performance is caused by SPEAR failing to recover one of the directions $\overline{q}_i$. We will clarify these points and provide further visualizations in the next version of the paper.
>
> **Q5: What is SPEAR’s performance under rigorous DPSGD guarantees?**
> We want to clarify that the results in Appendix E.6 were obtained by adding Gaussian noise to the gradient, as a viability test of SPEAR’s robustness against noisy gradients. Specifically, we did not do the gradient clipping necessary for DPSGD. We will clarify this in the paper and also provide new experiments, showing SPEAR’s results in the DPSGD setting alongside the corresponding  $(\epsilon,\delta)$ guarantee.

---

> ### Comment · Reviewer_38cL · 2024-08-12
>
> I did run the code provided by the author again, and the reconstruction results are fantastic. I am willing to raise my score since the authors addressed all the questions I asked. I believe this method is ground-breaking for gradient inversion attacks in general.  Other recent works tend to use diffusion/GAN to enhance their reconstruction results, but this work perfectly reconstructs the linear layer's result and demonstrates the possible vulnerability of LL in many NN architectures.
>
>
>  I just had a few detailed questions regarding implementation:
>
> 1. For VGG-16, are you running at linear shape = 1 1 4096 layer?
> 2. for implementation part, could you put Q_opt sperate only for extra evaluation function? I was confused and first thought you directly used GT in the reconstruction process.

---

> > ### Author Response · Authors · 2024-08-14
> >
> > We deeply appreciate the reviewer’s effort in reviewing our paper and code. We are thankful for their insightful questions, the
> > answers of which we will incorporate into the next revision of our paper. Below we respond to their additional questions above:
> >
> > **Q1:  For VGG-16, what is the shape of the attacked linear layer?**
> >
> > We use the standard ImageNet VGG-16 from torchvision, which uses AdaptiveAvgPool2d(7, 7), resulting in 25088 input features for the attacked linear layer. The attacked linear layer has 4096 output neurons. We will clarify this.
> >
> > **Q2: Computing $Q_\text{opt}$ in the main file of the attack is confusing**
> >
> > We agree with the reviewer that computing $Q_\text{opt}$, as one of the first things we do in our main can give the wrong impression we use the ground truth $Q$ in our reconstruction. We emphasize this is **not the case** as the reviewer correctly noticed, and we apologize for the confusion this has caused. We compute $Q_\text{opt}$ so that our sampling function *getQBarUniqueCol*, based on Theorem 3.3, in sparse\_gradient\_reconstruction.py can display how many directions were recovered correctly in real-time. This is incredibly valuable for debugging and monitoring the progress of our method. $Q_\text{opt}$ is also used to report the success of the different filtering stages of SPEAR described in Section 4 of the paper. This is done in the function *summarizeQ* in sparse\_gradient\_reconstruction.py, and it is useful for both debugging and for creating Figure 6 in Appendix E.3. We reiterate that $Q_\text{opt}$ is not used for any other purpose in our code. Before making our code public, we will do our best to restructure the code such that $Q_\text{opt}$ is only used at the end of the reconstruction for evaluation to avoid confusion.

---

### Official Review · Reviewer_Brdw · 2024-07-13

**Soundness:** 4
**Presentation:** 4
**Contribution:** 4
**Rating:** 7
**Confidence:** 3

**Summary:**

The paper presents SPEAR, a novel algorithm for exact gradient inversion of batches in federated learning. Unlike previous methods, SPEAR achieves exact reconstruction for larger batches by leveraging low-rank structure and ReLU-induced sparsity in gradients. The authors provide a theoretical foundation and an efficient GPU implementation, demonstrating the effectiveness of their approach on high-dimensional datasets and large networks.

**Strengths:**

1. The paper introduces a significant advancement in gradient inversion attacks by enabling exact reconstruction for batch sizes greater than one.
2. The authors provide a strong theoretical basis for their method, including proofs of the low-rank nature of gradients and the sparsity induced by ReLU activations.

**Weaknesses:**

1. The theoretical analysis only considers the ReLu and fully connected layer.
2. The method only seems to be effective for a small batch size, even though it has already been a big progress.

**Questions:**

SPEAR is effective for a moderate batch size of b < 25. What are the possible reasons to prevent the batch size to become larger?

**Limitations:**

See weakness.

---

> ### Author Rebuttal · Authors · 2024-08-07
>
> We thank the reviewer for the very positive review and are excited that the reviewer finds that SPEAR marks a significant advancement for gradient inversion. We are also happy to read that the reviewer attests that our method builds on a strong theoretical basis and credits our soundness, presentation and contribution. We answer all of the reviewer's questions in the general response:
>
> **Q1: SPEAR is effective for a moderate batch size of b < 25. What is preventing SPEAR from scaling beyond this?**
>
> Please see our detailed reply in the main response (Q1).
>
> **Q2: Can SPEAR handle activations beyond ReLU?**
>
> Please see our detailed reply in the main response (Q2).
>
> **Q3: Can SPEAR handle layers other than fully connected layers?**
>
> Please see our detailed reply in the main response (Q3).
>
> We are happy to discuss any remaining questions or follow-up questions the reviewer might have.

---

### Author Rebuttal · Authors · 2024-08-07

We thank the reviewers for their positive and helpful feedback. We are encouraged they consider our work to mark a significant advancement (Brdw), provide valuable insight (LPF2, Xnvh), our theory to be well founded (Brdw, 38cL), our presentation to be good (Brdw, LPF2, Xnvh) and our reconstruction results to look really good (38cL, Xnvh). Based on the reviewers' suggestions, we conducted a range of additional experiments, reporting results in the attached PDF. Below, we address the topics we found to be shared amongst reviewers or ones we believe are of major importance.

**Q1: What prevents SPEAR from scaling to larger batch sizes, and how can this limitation be overcome? (Reviewers Brdw, 38cL, LPF2, Xnvh)**
First, we would like to highlight that in the common regime where the batch size is small in comparison to the dimensions of the linear layer, we prove that w.h.p. the input information is losslessly represented in the gradient (see Figure 3). SPEAR is not failing for large batch sizes because the relevant information can not be retrieved but solely because we set a timeout on sampling as exponentially many samples would be required. A theoretical analysis of this is provided in Lemma 5.2.

This exponential sampling complexity can be addressed by combining SPEAR with an approximate reconstruction method to get a prior on which submatrices $L_A$ satisfy the conditions of Theorem 3.3, i.e., have corresponding matrices $A$ containing a 0-column. Using approximate, optimization-based input reconstruction techniques we can estimate the pre-activation values $Z$ and thus positions of $0$s in $\tfrac{\partial \mathcal{L}}{\partial Z}$. We can now use the estimated locations of the 0 entries in $\tfrac{\partial \mathcal{L}}{\partial Z}$ to directly sample $L_A$ matrices for which Theorem 3.3’s prerequisites are likely satisfied, drastically speeding up SPEAR’s information retrieval.

We confirm the effectiveness of this approach in a preliminary study, shown in Table 3 in the rebuttal PDF, allowing SPEAR to scale to batch sizes beyond 100. Specifically, we use the modern version of Geiping et al. [1] to get priors on $Z$ and considered the most negative entries of the approximated $Z$ to be the ones with the highest likelihood of actually being negative. We consider a network with $m=2000$ and 6 layers, and batches of 100 TinyImageNet images. There, Geiping has a PSNR of 32.8 while SPEAR reconstructs 6 out of 10 random batches exactly (PSNR of 120) and reconstructs 98 or 99 directions correctly on the remaining four batches, leading to an overall PSNR of 81.5. These results confirm that SPEAR can be effectively scaled to larger batches, with further optimizations being left to future work.

**Q2: The work is restricted to ReLU activations. Can you handle other activations, including sigmoid, leakyReLU, tanh, etc? (Reviewers Brdw, Xnvh)**

In this work, we focus on ReLU activations as they are the most relevant activation function in practice. However, more broadly, the core of our method is to utilize low rankness and priors on the activation function in order to not need priors on the client input data. While the low-rankness can directly be transferred to other activations, we believe extending the priors to other activations makes for an interesting future work item. We emphasize that SPEAR is the first exact method to reconstruct network inputs from gradients originating from batches with a batch size greater than 1.

**Q3: Is SPEAR applicable to architectures that do not have a fully connected layer immediately following their input? (Reviewers Brdw, 38cL, LPF2)**

Yes, it is. While the evaluation primarily focuses on recovering inputs to the first FC layer of a fully connected network with ReLU activations, our method allows the recovery of inputs of any FC layer preceded or followed by a ReLU activation regardless of the architecture used, as emphasized in Section 3.2. This is demonstrated in Table 2 in the rebuttal PDF, where we successfully recover the inputs to all layers followed by a ReLU in a $L=6$ layer FC network with width $m=400$. While these intermediate layer inputs may not violate the client privacy as severely as directly obtaining the user data exactly, the authors of [9] demonstrate that recovering individual batch inputs to intermediate layers can further boost the performance of optimization-based gradient leakage attacks and poses additional privacy issues for the clients.

Crucially, [9] makes this observation in the context of convolutional neural networks (CNNs). We also experimented with recovering the inputs to the FC layers in CNNs. Specifically, we attacked the first FC layer of a VGG16 network for ImageNet batches of size $b=20$. We present the results in Table 1 in the rebuttal PDF. Our results demonstrate that SPEAR recovers 100% of the individual linear layer inputs up to numerical precision. Thus, we are confident that SPEAR can improve optimization-based gradient leakage attacks for CNNs. We will add this to the paper.

Apart from CNNs, FC layers are the predominant layers for other domains, such as tabular data, where we significantly outperform the current state-of-the-art (Table 2, main paper). Further, for transformers, there is very promising follow-up work leveraging SPEAR’s insights to achieve state-of-the-art reconstruction of text by a wide margin. We have contacted the chair to share an anonymized copy of the follow-up work with you. All of this confirms the generality and applicability of SPEAR, as well as its significant impact on the field of gradient leakage.

**Conclusion**
We hope to have been able to address the reviewers’ questions and look forward to the discussion period.

---

### Decision · Program_Chairs · 2024-09-25

**Decision:**

Accept (poster)

**Comment:**

This paper focuses on proposing inversion attacks on federated learning methods that can reconstruct this data from the shared gradients. Specifically, the authors propose SPEAR which is a novel algorithm for exact gradient inversion of batches in federated learning. Unlike previous methods, SPEAR achieves exact reconstruction for larger batches by leveraging low-rank structure and ReLU-induced sparsity in gradients. The authors further provide a theoretical foundation and an efficient GPU implementation, demonstrating the effectiveness of their approach on high-dimensional datasets and large networks.

After the rebuttal/discussion period, all the reviewers are now of the opinion that the paper provides a set of novel contributions that are carefully evaluated. Hence, I would like to recommend the paper to be accepted. I thank the authors for their great efforts in addressing all the concerns. I would further like to suggest that the authors revise the manuscript carefully to address all the comments.